# Carbon dioxide utilization in concrete curing or mixing might not produce a net climate benefit

Dwarakanath Ravikumar[1,2 ✉], Duo Zhang[3], Gregory Keoleian [1], Shelie Miller [1], Volker Sick [4] & Victor Li[3]

Carbon capture and utilization for concrete production (CCU concrete) is estimated to sequester 0.1 to 1.4 gigatons of carbon dioxide ($CO_2$) by 2050. However, existing estimates do not account for the $CO_2$ impact from the capture, transport and utilization of $CO_2$, change in compressive strength in CCU concrete and uncertainty and variability in CCU concrete production processes. By accounting for these factors, we determine the net $CO_2$ benefit when CCU concrete produced from $CO_2$ curing and mixing substitutes for conventional concrete. The results demonstrate a higher likelihood of the net $CO_2$ benefit of CCU concrete being negative i.e. there is a net increase in $CO_2$ in 56 to 68 of 99 published experimental datasets depending on the $CO_2$ source. Ensuring an increase in compressive strength from $CO_2$ curing and mixing and decreasing the electricity used in $CO_2$ curing are promising strategies to increase the net $CO_2$ benefit from CCU concrete.

[1] Center for Sustainable Systems (CSS), School for Environment and Sustainability (SEAS), University of Michigan, Ann Arbor, MI, USA. [2] National Renewable Energy Laboratory (NREL), Golden, CO, USA. [3] Department of Civil and Environmental Engineering, University of Michigan, Ann Arbor, MI, USA. [4] Department of Mechanical Engineering, University of Michigan, Ann Arbor, MI, USA. ✉email: dwarak.ravikumar@nrel.gov

The capture and utilization of carbon dioxide ($CO_2$) to produce economically viable products offers the twin benefit of mitigating climate change and generating economically viable products[1]. Among the portfolio of products that can potentially utilize $CO_2$, concrete offers several advantages including (i) a thermodynamically favorable reaction mechanism[2,3] to sequester the $CO_2$ as calcium or magnesium carbonates[4,5] in ordinary Portland cement (OPC)[6,7] (ii) the long term $CO_2$ sequestration in the form of stable carbonate beyond the life-time of the infrastructure (>60 years) (iii) the significant sequestration potential due to the overall and expected growth in global cement production to meet increasing demand[8]. Concrete along with aggregates and chemicals/fuels are end-products with the potential to sequester the maximum quantity of $CO_2$ (in gigatons)[1,3].

Multiple emerging approaches such as carbonation of recycled concrete aggregates[9], $CO_2$ sequestration in alternative MgO based binders[10], $CO_2$ mineralization in industrial waste-derived aggregates and fillers[11,12], and $CO_2$ dissolution in mixing water[13,14] have been investigated for $CO_2$ utilization in concrete. However, this study focuses on the two approaches of $CO_2$ mixing and $CO_2$ curing as they are more extensively analyzed and applied for $CO_2$ utilization in concrete (Supplementary information (SI) Section 2). In $CO_2$ mixing, high-purity $CO_2$ is injected into fresh concrete during batching and mixing. The $CO_2$ binds to the calcium silicate clinker in OPC to form nano-scale $CaCO_3$ particles[15,16]. In $CO_2$ curing, $CO_2$ is utilized as a curing agent[5] to accelerate precast concrete fabrication. A review of $CO_2$ curing and mixing studies reveals that the $CO_2$ uptake potential in $CO_2$ curing of precast concrete applications is significantly higher than in $CO_2$ mixing (Supplementary Fig. 6).

A common assumption motivating research and commercial interests in CCU concrete is that the $CO_2$ uptake during curing and mixing[17–20] of CCU concrete lowers the $CO_2$ burden of concrete production. Estimates show that 0.1–1.4 gigatons of $CO_2$ can be utilized in concrete by 2050[1,3]. However, a literature review (SI Section 13) demonstrates these estimates are not based on a comprehensive assessment that accounts for the change in compressive strength of concrete from $CO_2$ utilization; the $CO_2$ impact of capturing, transporting and utilizing $CO_2$; the $CO_2$ emissions from compensating for the energy penalty of $CO_2$ capture and producing supplementary cementitious materials (SCM), which are by-products of coal electricity and pig iron production; the uncertainty and variability in inventory data and process parameters; and may not always be based on primary experimental data, which is required for a robust life cycle $CO_2$ assessment.

$CO_2$ curing can decrease the compressive strength of CCU concrete when compared to conventional concrete. For example, a review of 99 experimental datasets from existing literature shows that CCU concrete has a lower compressive strength than conventional concrete in 31 datasets (SI Section 2 Supplementary Fig. 3). In such cases, CCU concrete would require a greater amount of OPC than conventional concrete to produce the same compressive strength. OPC production is a major source of $CO_2$ emissions. Therefore, increased OPC content in a concrete formulation leads to an increase in $CO_2$ emissions from upstream cement production processes, which may outweigh the benefit of the $CO_2$ captured and used in concrete production.

In addition, the $CO_2$ impact of CCU concrete can be difficult to generalize due to the lack of consistency in the boundaries and scope of analysis. For example, the energy associated with the capture and transport of $CO_2$ is included in certain studies[21] while being excluded from others[20,22,23]. Moreover, the uncertainty and variability in data and process parameters, which is typical in the early stages of R&D, impacts the environmental assessment of emerging technologies such as CCU concrete[24–28].

Life cycle assessments (LCA) of CCU concrete rely on point values for process parameters rather than parameter distributions that provide a more realistic representation of uncertainty and variability[16,21]. The failure to account for uncertainty in the early stages of technology development can hinder research efforts to address hotspots and increase the $CO_2$ benefit from CCU concrete[25,26]. An uncertainty assessment in the early stages of technology development can determine process parameters and inventory items that are the most significant contributors to the $CO_2$ burden of CCU concrete and, thereby, help identify research strategies that are most effective in addressing the hotspots.

To address these issues, we review 99 datasets from 19 publications to determine the range of potential net $CO_2$ benefit associated with CCU concrete. The net $CO_2$ benefit is defined as the difference between the lifecycle $CO_2$ impact of producing conventional concrete and producing CCU concrete though $CO_2$ curing or $CO_2$ mixing. The net $CO_2$ benefit accounts for the life cycle $CO_2$ impact of the 13 upstream processes to capture, transport and utilize $CO_2$ and produce and transport the materials used in concrete. The net $CO_2$ benefit accounts for any changes in compressive strength when CCU concrete is produced though $CO_2$ curing or $CO_2$ mixing. We conduct a sensitivity analysis consisting of a scatter plot analysis and moment independent sensitivity analysis[25,29,30] to determine the key processes with the most significant influence on the net $CO_2$ benefit.

## Results and discussion

**Illustrative example: net $CO_2$ benefit for dataset 1**. We illustrate the interpretation of the results using a single dataset, which helps better understand the findings across the 99 datasets. Consider the dataset of conventional and CCU concrete production in the A1 scenario reported in ref. [31] wherein $CO_2$ is used for curing and OPC is used as binder. Based on the inventory requirements presented in the dataset, the $CO_2$ emissions for the 13 processes (Fig. 1 and Supplementary Table 1), the total life cycle $CO_2$ emissions from producing CCU concrete ($TOT_{CCU}$, Eq. 1) and conventional concrete ($TOT_{Conv}$, Eq. 5), and the net $CO_2$ benefit (Eq. 6) are stochastically determined in 10,000 Monte Carlo runs.

The distribution in Fig. 2a quantifies the likelihood of the net $CO_2$ benefit in dataset 1 being positive (right of the $y$-axis) in 10,000 Monte Carlo runs. If the net $CO_2$ benefit is positive, then the total life cycle $CO_2$ emissions from producing CCU concrete ($TOT_{CCU}$) is lower than conventional concrete ($TOT_{Conv}$). For dataset 1 the likelihood is 0%, which signifies that $TOT_{CCU}$ is greater than $TOT_{Conv}$ in all of the 10,000 Monte Carlo runs. Alternately, the distribution for the net $CO_2$ benefit is always negative which indicates that, on a life cycle basis, producing CCU concrete is more $CO_2$ intensive than conventional concrete in the 10,000 Monte Carlo runs.

The scatter plot analysis of the 10,000 Monte Carlo runs (Fig. 2b) determines the key drivers in the relationship between the difference in the $CO_2$ emissions from the 13 conventional concrete ($P_{Conv}$) and CCU concrete ($P_{CCU}$) processes and the net $CO_2$ benefit. The scatter plot shows the 10,000 net $CO_2$ benefit values on the ordinate and the 10,000 values of the difference between the $CO_2$ emissions from the 13 processes on the abscissa. A visual inspection of the scatter plot reveals a difference in the slopes of the scattered points. A higher slope indicates higher sensitivity of the net $CO_2$ benefit to the contributing process. The scatter plot for dataset 1 (Fig. 2b) demonstrates that the difference in the $CO_2$ emissions from cement used (P1:OPC) has a significant slope. As the difference in $CO_2$ emissions from OPC production are scattered in the lower left quadrant, the $CO_2$ emissions from OPC production for CCU concrete is greater than

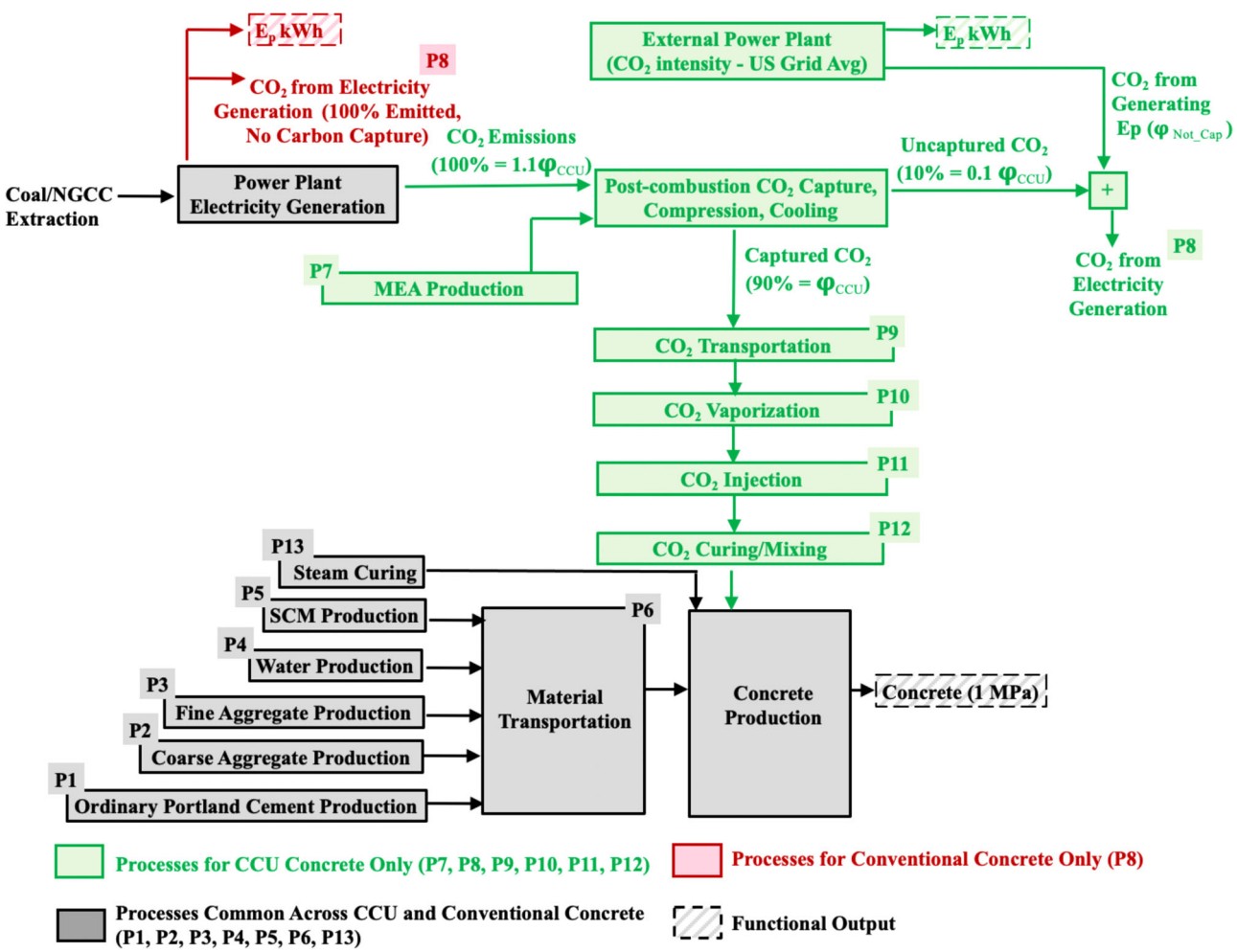

**Fig. 1 The system boundary diagram depicting the 13 processes, which were accounted for when determining the net CO$_2$ benefit of CCU concrete.** The processes required to produce CCU concrete are highlighted in gray and green. The processes required to manufacture conventional concrete are highlighted in gray and red. The CO$_2$ emissions, which is utilized in the curing or mixing of CCU concrete ($\varphi_{CCU}$ in kg), is captured from a power plant. The energy penalty from capturing $\varphi_{CCU}$ (E$_p$ kWh) is compensated by an external power plant. When conventional concrete is produced, there is no carbon capture during electricity generation and the CO$_2$ from generating E$_p$ in the power plant is completely emitted. The functional unit—1 m$^3$ of concrete with 1 MPa strength and E$_p$ kWh of electricity—is common across the CCU and conventional concrete production pathways. The CO$_2$ emissions from each of the CCU and conventional concrete production process is quantified in Eqs. 1 and 5.

conventional concrete. Therefore, for dataset 1, the difference between the CO$_2$ emissions from OPC production is the most important reason for TOT$_{CCU}$ being greater than TOT$_{Conv}$ (i.e., the net CO$_2$ benefit being negative).

This finding is confirmed by a moment independent sensitivity analysis[25,29,30], which determines δ indices for each of the 13 processes contributing to the net CO$_2$ benefit (Fig. 2c). The δ index is a measure of the contribution from the difference in the CO$_2$ emissions from a conventional and CCU concrete production process to the probability distribution function of the net CO$_2$ benefit. The process with a greater δ index value has a greater contribution to the net CO$_2$ benefit than a parameter with a lower δ index value. By convention, if the δ value is negative then the CO$_2$ emissions from the process are greater in CCU concrete production than in conventional concrete production. The difference in the CO$_2$ emissions from OPC production has the highest δ value and is, therefore, the most significant contributor to the net CO$_2$ benefit. The negative value indicates that the CO$_2$ emissions from OPC production are greater in CCU concrete than in conventional concrete. The finding from the sensitivity analysis for dataset 1 can be attributed to the compressive strength of CCU concrete (16–17.4 MPa) being

lower than conventional concrete (18–18.6 MPa). The mean compressive strength of CCU concrete (16.7 MPa) is 9% lower than that of conventional concrete (18.3 MPa). This implies that in dataset 1 a greater mass of OPC is produced for CCU concrete to achieve the same compressive strength as conventional concrete. In dataset 1, the OPC produced per MPa for CCU concrete is 24.8 kg/MPa and for conventional concrete is 22.6 kg/MPa.

Using a mean value of 0.948 kg CO$_2$/kg OPC for the life cycle CO$_2$ footprint of OPC (Supplementary Table 2), the CO$_2$ emissions from OPC production for CCU concrete is 23.5 kg CO$_2$/MPa and for conventional concrete is 21.4 kg CO$_2$/MPa. Therefore, the difference between the CO$_2$ emissions from OPC production for CCU and conventional concrete is 2.1 kg CO$_2$/MPa, which is greater than the CO$_2$ utilized in CCU concrete (1.1 kg CO$_2$/MPa, SI Section 1). As a result, the life cycle CO$_2$ emissions from the increased OPC production in CCU concrete is greater than the CO$_2$ utilized for curing the CCU concrete.

**Increased binder use undermines CO$_2$ benefit of CCU concrete.** We extend the analysis conducted for dataset 1 to the remaining 98 datasets (Fig. 3). The 99 datasets are divided into 4 categories.

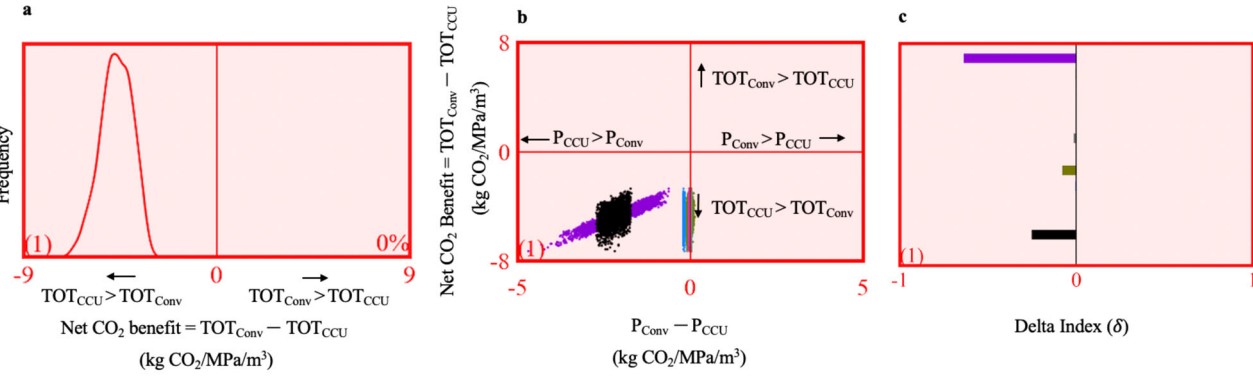

**Fig. 2 Illustrative example to explain the interpretation of the results. a** The distribution of the net $CO_2$ benefit of CCU concrete in 10,000 Monte Carlo runs. The net $CO_2$ benefit is the difference between the total $CO_2$ emissions from producing conventional concrete ($TOT_{Conv}$) and CCU concrete ($TOT_{CCU}$) **b** the scatter plot demonstrating the sensitivity of the net $CO_2$ benefit to the difference in the $CO_2$ emissions from the 13 contributing processes for conventional concrete ($P_{Conv}$) and CCU concrete ($P_{CCU}$). Positive values on the y-axis indicates that $TOT_{Conv} > TOT_{CCU}$ (upper right and upper left quadrants), negative values on the y-axis indicate that $TOT_{CCU} > TOT_{Conv}$ (lower right and lower left quadrants), positive values on the x-axis indicate that $P_{Conv} > P_{CCU}$ (upper right and lower right quadrants), negative values on the x-axis indicate that $P_{CCU} > P_{Conv}$ (upper left and lower left quadrants) **c** delta indices to determine the sensitivity of the net $CO_2$ benefit to the difference between the $CO_2$ emissions from $P_{Conv}$ and $P_{CCU}$. The number in parenthesis represents the dataset number (from the literature review) for which the results are determined. The red background in (**a–c**) signifies that the net $CO_2$ benefit is negative (i.e., $TOT_{CCU}$ is greater than $TOT_{Conv}$) in at least 5000 out of the 10,000 Monte Carlo runs (i.e., likelihood greater than 50%). If the background is green, then it signifies that the net $CO_2$ benefit is positive with a likelihood greater than 50%.

Category 1 has 50 datasets wherein $CO_2$ is used for curing and the binder consists of only OPC. Category 2 has 20 datasets wherein $CO_2$ is used for curing and the binder consists of a mix of OPC and supplementary cementitious materials (SCM). Category 3 has 8 datasets wherein $CO_2$ is used for mixing and the binder consists of only OPC. Category 4 has 21 datasets wherein $CO_2$ is used for mixing and the binder consists of a mix of OPC and SCM.

A visual inspection of the slopes in the scatter plot in Fig. 4 and the δ indices in Fig. 5 reveal that the net $CO_2$ benefit is most sensitive to the amount of OPC produced and used in the design mix (e.g., P1 in datasets 1, 2, 3, and 4 in Fig. 5), the energy used for $CO_2$ curing (e.g., P12 in datasets 21, 22, and 23 in Fig. 5) and SCM produced and used (e.g., P5 in datasets 51, 52, 53, and 54 in Fig. 5).

The plots with the red background in Figs. 3, 4, and 5 demonstrate that the net $CO_2$ benefit is negative (i.e., CCU concrete has higher life cycle $CO_2$ emissions than conventional concrete) with at least a 50% likelihood in 56 out of the 99 datasets. The compressive strength in CCU concrete decreases due to $CO_2$ curing when compared to conventionally cured concrete. The OPC and SCM consumed to produce the same compressive strength is greater in CCU concrete than in conventional concrete. Therefore, the results demonstrate the $CO_2$ burden of increased OPC and SCM consumption for CCU concrete outweighs the benefit of the $CO_2$ that is captured and used in CCU curing. Additionally, in category 1 datasets, the electricity use in the $CO_2$ curing process is the second key contributor to the increase in the total life cycle $CO_2$ emissions from CCU concrete (e.g., datasets 46, 47, and 48 in Fig. 5).

The results in Fig. 5 also demonstrate that the life cycle $CO_2$ emissions from capturing, compressing, transporting and vaporizing $CO_2$, and the $CO_2$ emissions from producing fine aggregate, coarse aggregate, water and steam curing are not significant contributors to the net $CO_2$ benefit. This can be attributed to the mass of $CO_2$ utilized in concrete being lower than the mass of the cement and coarse and fine aggregate (Supplementary Fig. 1) and the life cycle $CO_2$ intensity of coarse and fine aggregate being significantly lower than cement (Supplementary Table 2).

To investigate the change in results when $CO_2$ intensity of OPC production decreases, we determine the net $CO_2$ benefit of CCU concrete when $CO_2$ is captured from a cement plant (SI Section 11). The results show that CCU concrete has higher life cycle $CO_2$ emissions than conventional concrete (i.e., negative $CO_2$ benefit) in 44 out of the 99 datasets (Supplementary Fig. S4 in SI Section 11) when compared to 56 out of the 99 datasets in the baseline scenario (Fig. 5). Therefore, when $CO_2$ is captured from a cement plant, there is a lower likelihood of CCU concrete producing a negative net $CO_2$ benefit than when $CO_2$ is captured from a power plant. The difference in the results can be attributed to the reduced $CO_2$ intensity of cement production due to $CO_2$ capture at the cement plant.

**Results across $CO_2$ use, binder type, and allocation method.** Fig. 6 summarizes the results depending on whether $CO_2$ is used for curing or mixing, SCM is used as a binder material or not, and the type of allocation method used to determine the $CO_2$ emissions from producing the SCM material. Two types of SCMs are used in concrete production—ground granulated blast furnace slag and fly ash. Slag is a by-product of pig-iron production and fly ash is a by-product of electricity generation in coal power plants. As a result, we need a method to allocate total $CO_2$ emissions between slag and pig-iron and between fly-ash and coal electricity. We use three methods—system expansion (SE), mass-based (MA), and economic value-based allocation (EA)—to allocate and account for the life cycle $CO_2$ emissions from producing the SCMs (process P5, SI Sections 4, 5, and 6).

The overall results in Fig. 6 demonstrate that in 36 (EA in "Overall") to 43 (SE in "Overall") of the 99 datasets reported in the literature, CCU concrete production has a lower life cycle $CO_2$ emission than conventional concrete production. In these cases, CCU concrete substituting conventional concrete lowers $CO_2$ emissions. Negative $CO_2$ net benefit values are obtained in the remaining 56 to 63 datasets. A similar analysis for $CO_2$ capture from a NGCC power plant shows that the net $CO_2$ benefit is negative in 61, 65, and 68 of the 99 datasets when SE, MA, and

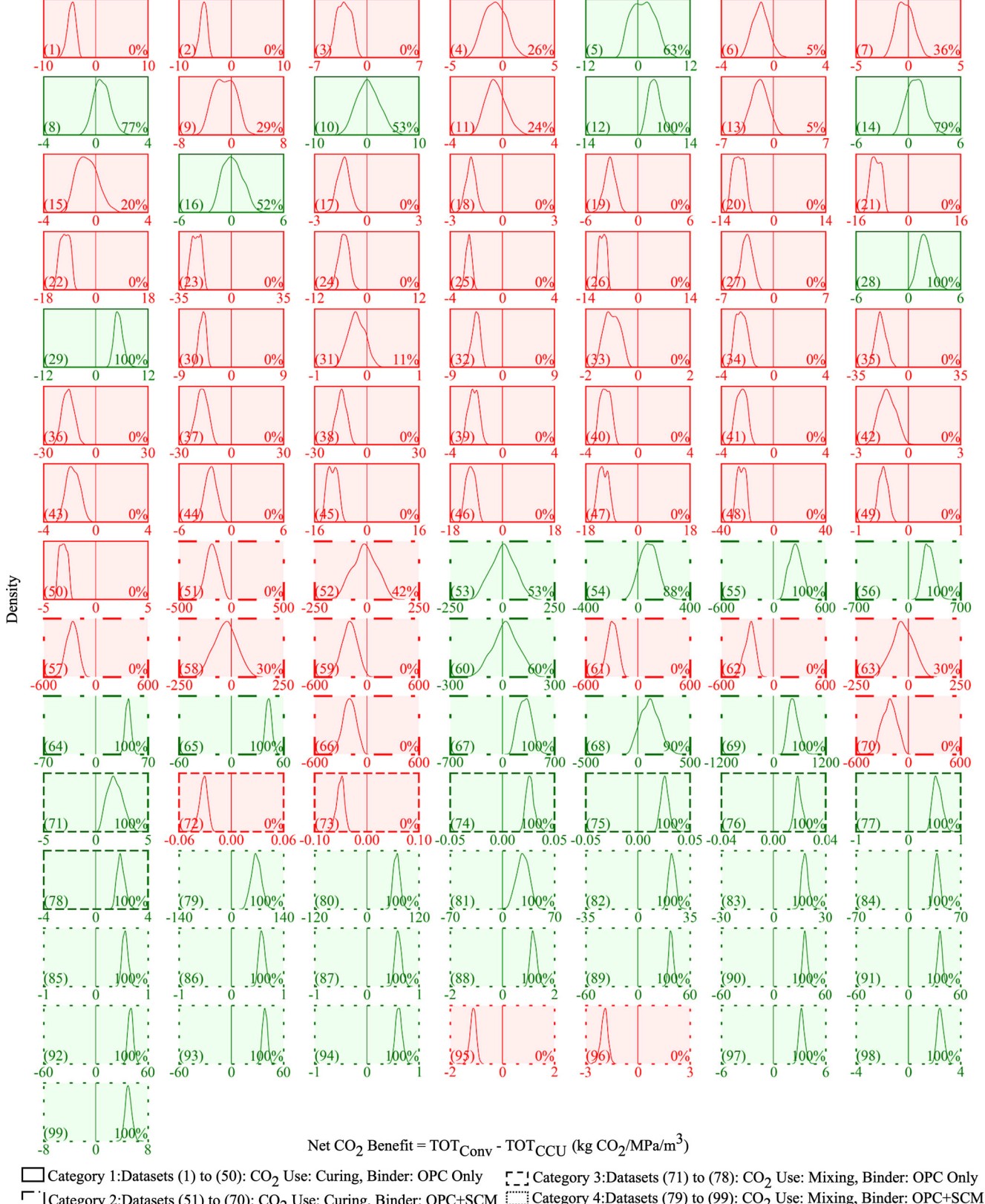

$$\text{Net CO}_2 \text{ Benefit} = \text{TOT}_{\text{Conv}} - \text{TOT}_{\text{CCU}} \; (\text{kg CO}_2/\text{MPa/m}^3)$$

☐ Category 1: Datasets (1) to (50): CO$_2$ Use: Curing, Binder: OPC Only ⊡ Category 3: Datasets (71) to (78): CO$_2$ Use: Mixing, Binder: OPC Only
⊟ Category 2: Datasets (51) to (70): CO$_2$ Use: Curing, Binder: OPC+SCM ⬚ Category 4: Datasets (79) to (99): CO$_2$ Use: Mixing, Binder: OPC+SCM

**Fig. 3 The net CO$_2$ benefit of CCU concrete production across 99 datasets.** The curve in each plot represents the distribution of the net CO$_2$ benefit, which is the difference between the total CO$_2$ emissions from producing conventional and CCU, across 10,000 Monte Carlo runs. When the net CO$_2$ benefit is negative in at least 5000 of the 10,000 Monte Carlo runs (50% likelihood), the background is red. If the background is green, then it signifies that the net CO$_2$ benefit is positive with a likelihood greater than 50%. The likelihood is presented as a percentage value in the lower right corner of the plot. The parenthesized value in the lower left corner is the dataset number.

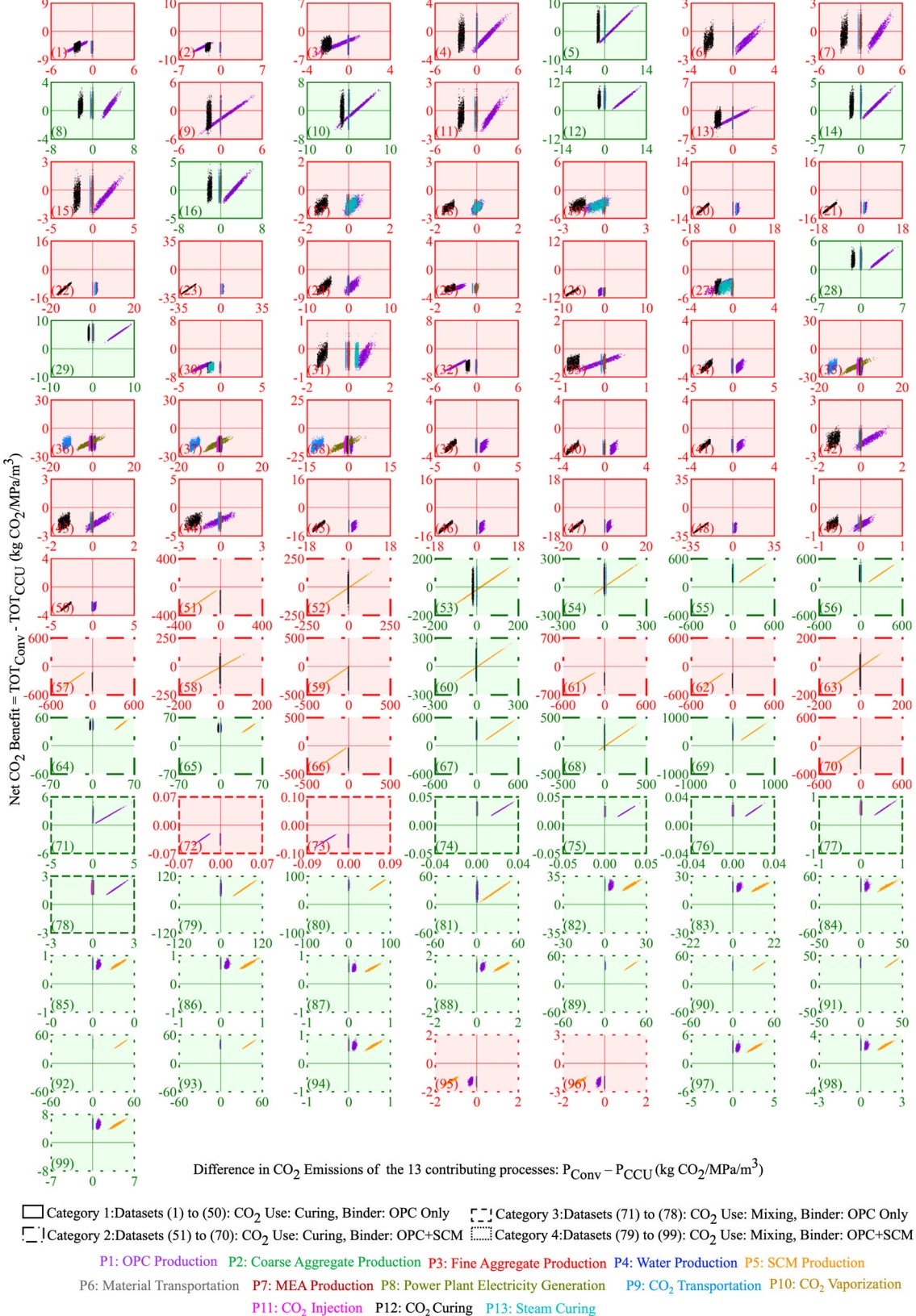

**Fig. 4 Scatter plot analysis to determine the impact of the 13 processes on the net CO₂ benefit across the 99 datasets.** The difference between the $CO_2$ emissions from the 13 processes of CCU and conventional concrete production is plotted on the x-axis and the net $CO_2$ benefit is plotted on the y-axis. The parenthesized value in the lower left corner is the dataset number. Higher resolution scatter plots for each of the 13 processes can be downloaded from SI Section 9 Supplementary Table 12.

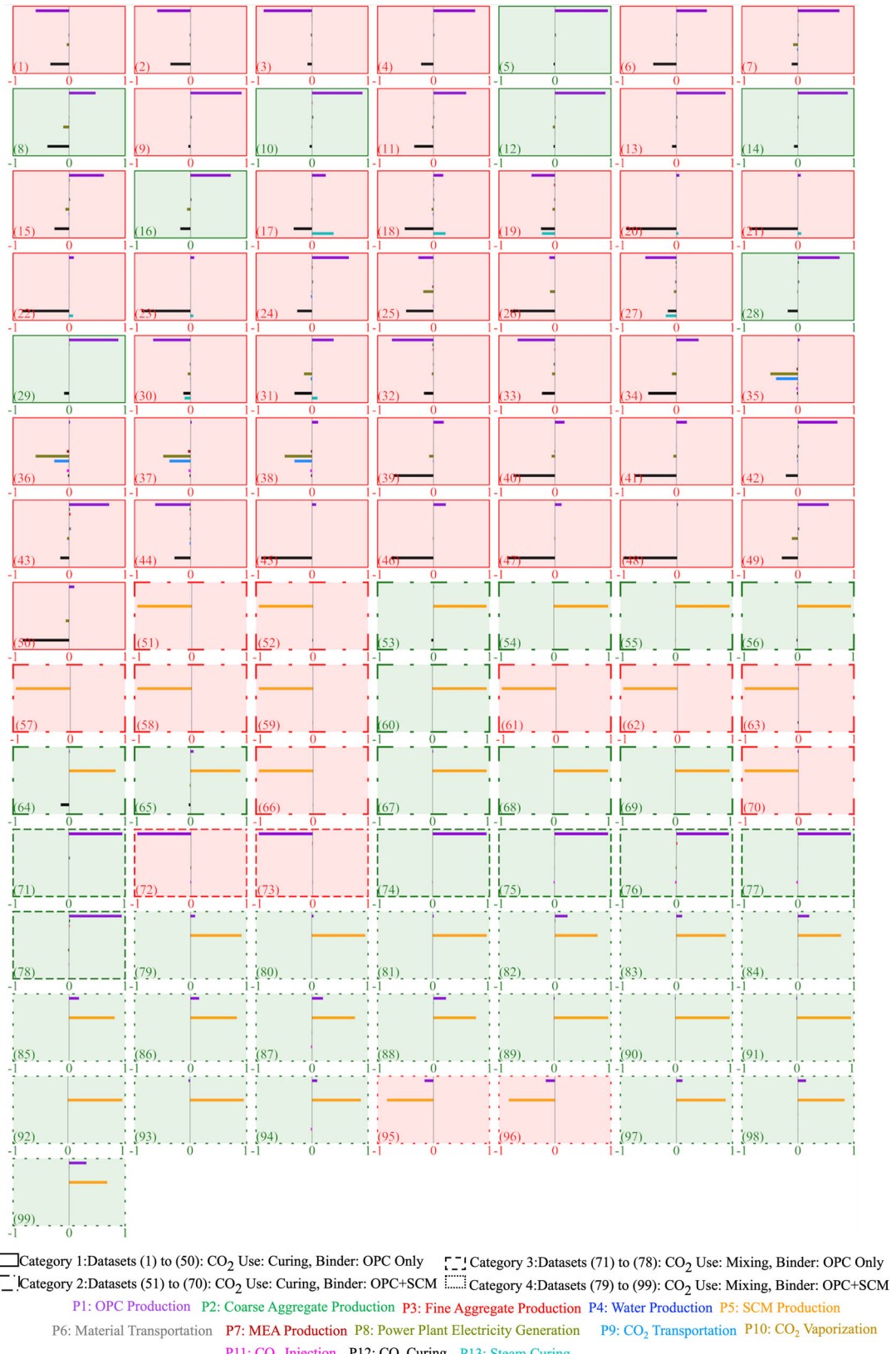

Fig. 5 δ indices quantifying influence of the difference between the $CO_2$ emissions from the 13 processes of CCU and conventional concrete production on the net $CO_2$ benefit. A process with a greater δ index value has a greater influence on the net $CO_2$ benefit than a process with a lower δ index value. The parenthesized value in the lower left corner is the dataset number.

Category 1:Datasets (1) to (50): $CO_2$ Use: Curing, Binder: OPC Only
Category 3:Datasets (71) to (78): $CO_2$ Use: Mixing, Binder: OPC Only
Category 2:Datasets (51) to (70): $CO_2$ Use: Curing, Binder: OPC+SCM
Category 4:Datasets (79) to (99): $CO_2$ Use: Mixing, Binder: OPC+SCM

P1: OPC Production   P2: Coarse Aggregate Production   P3: Fine Aggregate Production   P4: Water Production   P5: SCM Production
P6: Material Transportation   P7: MEA Production   P8: Power Plant Electricity Generation   P9: $CO_2$ Transportation   P10: $CO_2$ Vaporization
P11: $CO_2$ Injection   P12: $CO_2$ Curing   P13: Steam Curing

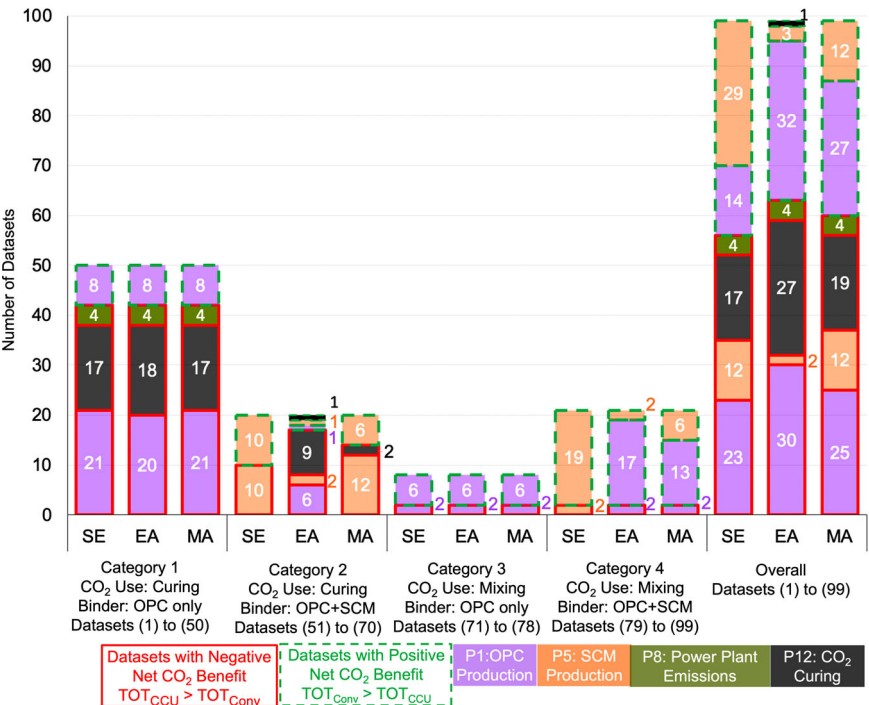

**Fig. 6 Results providing a break-up of the datasets with positive and negative net $CO_2$ benefit from CCU concrete and the most significant driver of the net $CO_2$ benefit of CCU concrete across category 1, category 2, category 3, and category 4 datasets (SI Section 2).** The results were determined using system expansion (SE), economic value-based (EA), and mass-based (MA) allocation to determine the $CO_2$ emissions from slag (co-product of iron ore production) and fly ash (co-product of coal electricity generation), which are used as SCM in category 2 and 4 datasets.

EA are used, respectively (SI Section 10). The overall results demonstrate that the $CO_2$ benefit of CCU concrete production is negative in 56–68 of the 99 datasets depending on whether $CO_2$ is captured from a coal or a NGCC power plant and when SE, MA or EA is used. As a result, there is a higher likelihood of the net $CO_2$ benefit of CCU concrete being negative. Consequently, not all CCU concretes can help realize sequestering 0.1 to 1.4 gigatons/year of $CO_2$ in concrete by 2050[1,3].

The results of the sensitivity analysis are similar across categories 1, 2, 3, and 4 as the production and use of binder material (OPC in categories 1 and 3 and OPC + SCM in Categories 2 and 4) has the most significant influence on the net $CO_2$ benefit.

The overall results on the number of datasets in which there is a positive net $CO_2$ benefit (i.e., CCU is less $CO_2$ intensive than conventional concrete) is not significantly impacted by the choice of system expansion SE, MA or EA. CCU is less $CO_2$ intensive than conventional concrete in a minimum of 36 datasets when EA is used and a maximum of 43 datasets when SE is used. However, for category 2 and 4 datasets, the choice of SE, MA or EA impacts the results from the δ indices sensitivity analysis. When SE or MA is used in category 2 datasets, SCM use is a key contributor to the net $CO_2$ benefit of CCU concrete. When EA is used in category 2 datasets, the $CO_2$ emissions from the $CO_2$ curing process is the key contributor. When SE is used in category 4 datasets, SCM use is a key contributor to the difference in the total $CO_2$ emissions between conventional and CCU concrete. When EA or MA is used in category 4 datasets, OPC use is the key contributor. However, the choice of SE, MA or EA is an artifact of the method of analysis—not actually changing real $CO_2$ emissions—and, therefore, is not within the purview of engineering strategies to improve processes and decrease the $CO_2$ emissions from producing CCU concrete.

The findings from this analysis are based on the design mixes, material usage, compressive strength, and parameters such as the

$CO_2$ curing duration, water to cement and SCM to cement ratios obtained from the 99 datasets (SI Section 2 Supplementary Figs. 1 to 5). It is important to note that the findings do not preclude future research (as discussed below) from optimizing the design mixes, curing processes and material properties to increase the net $CO_2$ benefit from CCU concrete.

**Strategies to improve the net $CO_2$ benefit of CCU concrete.** An R&D agenda focused on the following items, which are within the control of the CCU concrete production process, can increase the net $CO_2$ benefit.

(i)   Ensure increase in compressive strength from $CO_2$ curing: a key priority is to determine a $CO_2$ curing protocol that consistently increases the compressive strength of CCU concrete. An increase in compressive strength implies that a smaller quantity of carbon intensive binder material is used in CCU concrete to achieve the same compressive strength as conventional concrete (i.e., lower quantity of OPC or SCM is consumed on a kg per MPa basis). Fine tuning the curing process such as duration of the pre-hydration and post-carbonation water compensation are promising candidates to restore the reduction in 28-day compressive strength observed for $CO_2$-cured concrete[32–34]. For example, a longer duration of pre-hydration may enhance 28-day compressive strength but decreases $CO_2$ uptake at early age. Further investigations are needed for enhancing consistency of $CO_2$-cured concrete production and for implementing the laboratory strategies in field applications.

(ii)  Decrease the $CO_2$ emissions from the $CO_2$ curing process: electricity use, which is the key contributor to the $CO_2$ emitted during the $CO_2$ curing process, can be lowered by streamlining the curing process. Future research can investigate and standardize promising options, such as natural drying or waste heat drying for pre-curing of $CO_2$-

cured concrete[35] with the end goal of accelerating adoption in industry.

(iii) Improve understanding on the impact of $CO_2$ curing on durability: the findings of this research are based on the compressive strength property of CCU concrete, which is limited from a lifecycle perspective. Prior studies show that construction and repair frequencies are key drivers in determining concrete life cycle $CO_2$ impacts[36,37]. Therefore, the effect of CCU on concrete durability must be considered when analyzing life cycle $CO_2$ emissions. Preliminary lab scale studies demonstrate that $CO_2$ curing improves durability related parameters such as permeability, sorptivity and sulfate and acid resistance[22,38–40]. However, the variability in the curing conditions and the design mixes used in the studies should be accounted for to ensure that durability gains can be consistently realized when $CO_2$ curing of concrete is adopted at a commercial scale. Future work can prioritize standardizing the $CO_2$ curing protocol (e.g., the steam curing time, pre-hydration time, post-hydration time), and study the resulting durability impact on different design mixes (e.g., use of different SCMs), with the overall goal of identifying optimal curing conditions and design mixes to maximize durability. This applies to ready-mix concrete and general precast applications with end-products such as masonry units[32,35], pipes[22], and pavers[41]. In addition, $CO_2$ curing for reinforced concrete needs further investigation due to the possibility of increased risk of steel reinforcement corrosion led by concrete carbonation. Moreover, CCU can be potentially synergized with established strategies for concrete crack width control, e.g., engineered cementitious composites with microfiber reinforcement, to further promoting concrete durability[42,43].

The system boundary (Fig. 1) assumes that the $CO_2$ captured from the power plant is used for CCU concrete production without any intermediate storage. In practice, the total $CO_2$ captured from a power plant may be significantly greater than the maximum utilization capacity at a CCU concrete production plant. In such cases, the excess captured $CO_2$ may be temporarily stored for future utilization in CCU concrete production or routed towards other utilization pathways. Given that $CO_2$ utilization is an emerging field and in the early stages of commercialization, there is a lack of time-sensitive data on how the captured $CO_2$ feedstock is either temporarily stored or immediately allocated to other utilization pathways. As a result, a system boundary that incorporates the time-sensitive utilization of $CO_2$ captured from a power plant is beyond the scope of this work and is a topic for future research.

The transport of $CO_2$ through pipelines has a lower $CO_2$ impact than road-based transport using semi-trailer trucks[44], which is modeled in this analysis. To quantify the maximum possible gains from shifting to a less carbon intensive mode of $CO_2$ transport, we conduct a scenario analysis with the optimistic assumption that the $CO_2$ impact of $CO_2$ transportation is zero (SI Section 12). Despite this optimistic assumption of zero-carbon $CO_2$ transport, CCU concrete has a lower $CO_2$ impact than conventional concrete in 44 of the 99 datasets, which is similar to the 43 of the 99 datasets obtained in the baseline scenario (Supplementary Fig. 10 versus Fig. 5). As a result, a shift from road to pipeline based $CO_2$ transport will not impact the findings from this analysis.

This analysis focusses on the use of pure $CO_2$ and two approaches of $CO_2$ utilization—curing and mixing—as they are more extensively investigated (e.g., 99 datasets used in this study) than alternate approaches such as concrete curing with flue gas[21,45–47], carbonation of recycled concrete aggregates[9], $CO_2$ sequestration in alternative MgO based binders[10], and $CO_2$ dissolution in mixing water[13,14]. The increased availability of experimental data is necessary to robustly quantify the net $CO_2$ benefit of CCU concrete and account for the impact of data uncertainty and process variability on the results. For example, further experimental research can generate data on the variability in the compressive strength of flue-gas cured concrete properties when the 2-week curing time is reduced[47] and the $CO_2$ concentration in the flue gas is varied[45]. With increased availability of inventory and process data from future experimental research, the life cycle approach presented in this study can be extended to quantify the net $CO_2$ benefit of CCU concrete produced from flue gas and other alternate approaches of $CO_2$ utilization.

The impact of $CO_2$ curing or $CO_2$ mixing on the lifetime behaviors and geography specific factors are important practical considerations when using CCU concrete in commercial applications. The lifetime behaviors are impacted by the variations in the geographical sources of raw materials, mix type, product type and the service environment in which the concrete is deployed. For example, the impact of $CO_2$ curing on concrete lifespan would differ substantially between concrete with and without steel reinforcement, due to the heightened steel corrosion[48] caused by the $CO_2$-induced pH reduction[49]. Additionally, geography specific factors such as sulfate-rich soils[50], cold regions[39], or acidic environments[22] can impact the durability of CCU concrete. The findings of this study can be further complemented by future research which quantifies the impact of variations in the lifetime and the geography specific factors on the net $CO_2$ benefit of CCU concrete.

## Methods

**Literature review to categorize $CO_2$ use in concrete**. We conducted a literature review to obtain the 99 datasets from 19 studies presenting life cycle material and energy inventory data and process parameters for the production of CCU and conventional concrete. The literature review identified the 19 studies[16,19,22,23,31–33,35,38,40,51–59] as they were the only ones to report the following three items (i) the design mix consisting of the energy and material inventory required for the production of conventional and CCU concrete (SI Section 2). The energy and material inventory are required to determine the life cycle $CO_2$ impact of producing conventional and CCU concrete; (ii) the quantity of $CO_2$ used in mixing or curing of concrete. This is required to determine the life cycle $CO_2$ impact of capturing, transporting, and utilizing the $CO_2$ used in producing CCU concrete; and (iii) the compressive strength of CCU and conventional concrete at the end of 28 days, which helps account for the change in the material property between conventional and CCU concrete. The 28-day compressive strength is among the most widely used technical parameters for assessing concrete quality, categorizing concrete mix designs[60] and forms the basis for concrete structural design[61,62] and is, therefore, chosen as the functional property based on which conventional and CCU concrete are compared. Based on whether $CO_2$ is used in CCU concrete for curing or mixing and if SCM was used in the design mix, the 99 datasets were organized into four categories.

(i) Category 1: $CO_2$ is used in the curing of concrete and only OPC is used as the cementitious material in the design mix[22,31,33,38,40,56–59]. This category contains 50 datasets.

(ii) Category 2: $CO_2$ is used in the curing of concrete and a combination of OPC and SCM is used as the cementitious material in the design mix[23,32,35,55]. This category contains 20 datasets.

(iii) Category 3: $CO_2$ is used in the mixing of concrete and only OPC is used as the cementitious material in the design mix[16,19,51]. This category contains 8 datasets.

(iv) Category 4: $CO_2$ is used in the mixing of concrete and a combination of OPC and SCM is used as the cementitious material in the design mix[16,51–54]. This category contains 21 datasets.

SCM took the form of either ground granulated blast furnace slag, which is a by-product of the pig-iron production[63], or fly ash, which is a by-product of electricity generation in coal power plants.

**Functional Unit**. The use of $CO_2$ during mixing or curing changes the compressive strength of CCU concrete when compared to concrete produced through conventional mixing or curing. In addition, an energy penalty ($E_p$ kWh) is incurred for

CCU concrete in power plants due to the energy associated with capturing the $CO_2$, which is used in the curing or mixing of CCU concrete ($\varphi_{CCU}$, kg $CO_2$). $E_p$ is not incurred when conventional concrete is produced since there is no $CO_2$ capture. Therefore, the net $CO_2$ benefit of substituting CCU concrete for conventional concrete should account for the $CO_2$ impact from the change in the compressive strength and $E_p$, which is incurred in power plants only when $CO_2$ is captured.

As a result, we use a functional unit of concrete with 1 MPa compressive strength and 1 $m^3$ of volume and $E_p$ kWh of electricity.

The functional unit accounts for the change in compressive strength and ensures consistency by normalizing the materials and energy consumed for producing 1 $m^3$ of CCU and conventional concrete to 1 MPa of compressive strength. The inclusion of $E_p$ kWh of electricity in the functional unit accounts for the difference in $CO_2$ emissions from electricity generation without $CO_2$ capture in the conventional concrete pathway and with $CO_2$ capture in the CCU concrete pathway. $E_p$ is determined based on the mass of $CO_2$ captured from the power plant (Supplementary Table 1 Process 8).

**CCU concrete production—system boundary and $CO_2$ emissions.** The literature review revealed that the total life cycle $CO_2$ emissions from producing CCU concrete is the sum of the $CO_2$ emissions from 13 key processes required to capture, transport, and utilize $CO_2$ and produce the materials required in the design mix of concrete (Fig. 1).

The expression used to determine the total life cycle $CO_2$ emissions from producing CCU concrete based on the $CO_2$ emissions from the 13 processes is presented in Eq. 1. The 13 expressions within parenthesis in Eq. 1 correspond to the $CO_2$ emissions from the 13 processes (Fig. 1).

$$
\begin{aligned}
TOT_{CCU} = & \left(\varphi_C * C_{CCU}\right) + \left(\varphi_{CA} * CA_{CCU}\right) + \left(\varphi_{FA} * FA_{CCU}\right) + \left(\varphi_W * W_{CCU}\right) \\
& + \left(\varphi_{SCM} * SCM_{CCU}\right) + \left(D_M * \varphi_{TM} * M_{Conv}\right) \\
& + \left(\varphi_{CCU} * j_{MEA}\right) + \left(Alloc_{elec} * \varphi Not\ Cap + \varphi_{Avg} * E_p\right) \\
& + \left(\varphi_{CCU} * (1 + 2T_w) * D_{CO2} * \varphi_T\right) \\
& + \left(\varphi_{CCU} * \varphi_{Vap}\right) + \left(\varphi_{CCU} * \left(\varphi_{Inj} + (1 - \eta)\right)\right) \\
& + \left(\varphi_{CO2\_Cur}\right) + \left(\varphi_{Stm\_Cur}\right)
\end{aligned} \tag{1}
$$

Process 1 to 4—Ordinary Portland cement (C), coarse aggregate (CA), fine aggregate (FA), and water (W) production: The $CO_2$ impact is the product of (i) the life cycle $CO_2$ emissions from producing the material ($\varphi_C$, $\varphi_{FA}$, $\varphi_{CA}$ and $\varphi_W$ in kg $CO_2$/kg material) and (ii) and the mass of material used in the design mix normalized to the compressive strength of CCU concrete ($C_{CCU}$, $CA_{CCU}$, $FA_{CCU}$ and $W_{CCU}$ in kg material/MPa/$m^3$). The material used and the compressive strength are obtained from the literature review (SI Section 2) and $\varphi_C$, $\varphi_{FA}$, $\varphi_{CA}$, and $\varphi_W$ are obtained from the ecoinvent database (Supplementary Table 2).

Process 5—SCM production: $SCM_{CCU}$ represents the mass of SCM used in the design mix normalized to the compressive strength of CCU concrete (in kg material/MPa/$m^3$).

Slag and fly ash, which are co-products of iron-ore production and electricity generation from coal, are used as SCM in the design mix of concrete. Three methods—system expansion (SE), economic value-based allocation (EA) and mass-based allocation (MA)—are widely used in LCA to determine the $CO_2$ emissions of co-products being generated by a single system.

In SE, the $CO_2$ emissions from producing a required mass of slag is determined by expanding the system to include the production of a corresponding mass of iron-ore (based on a ratio of iron-ore to slag, SI Section 4). In the case of MA and EA, the total $CO_2$ emission from a process producing both iron-ore and slag is allocated between iron-ore and slag based on the mass and economic value of the co-products, respectively (SI Sections 5 and 6). To explore variability in the $CO_2$ emissions from CCU concrete production based on the allocation method, this analysis applies the three methods when determining the $CO_2$ emissions for slag and fly ash.

The $CO_2$ impact of slag ($\varphi SCM\_slag$ in kg $CO_2$/kg slag) is determined from Eq. 2

$$\varphi_{SCM\_slag} = Alloc_{slag} * 7.7 * \varphi_{IO} \tag{2}$$

The value of $Alloc_{slag}$ is 1, 0008. or 0.11 when SE, MA or EA is chosen, respectively (SI Sections 4, 5 and 6).

$\varphi_{IO}$ is the life cycle $CO_2$ emissions from producing 1 kg of iron ore and is 2.2 kg $CO_2$/kg iron ore (SI Section 4).

When fly ash is used as the SCM, the $CO_2$ impact per kg of fly ash ($\varphi_{SCM\_ash}$ in kg $CO_2$/kg fly ash) is determined from Eq. 3

$$\varphi_{SCM\_ash} = Alloc_{ash} * 22.7 * \varphi_{Elec\_Coal} * \alpha_{Cap} \tag{3}$$

The value of $Alloc_{ash}$ is 1, 0.02 or 0.06 when SE, MA or EA is chosen, respectively (SI Sections 4, 5 and 6). $\varphi_{Elec\_Coal}$, which is the life cycle $CO_2$ emission from producing 1 kWh of coal electricity, is 1.25 kg $CO_2$/kWh (SI Section 4). $\alpha_{Cap}$ is 0.1 if $CO_2$ is captured at a coal plant and used in CCU concrete production. $\alpha_{Cap}$ is 1 if there is no carbon capture at a coal plant i.e., when $CO_2$ is captured from a combined cycle natural gas plant and used in CCU concrete production.

Process 6—Material Transportation: The $CO_2$ emissions from material transport is the product of the 5 materials used in the design mix ($M_{CCU}$ in kg/MPa/$m^3$), the $CO_2$ intensity of the mode of transportation used ($\varphi_M$ in kg $CO_2$ per kg-km) and the distance over which the materials are transported ($D_M$ in km). $M_{CCU}$ represents $C_{CCU}$, $FA_{CCU}$, $CA_{CCU}$, $W_{CCU}$ and $SCM_{CCU}$ from processes 1 to 5. $D_M$ values for road, rail, ocean and barge transport are obtained from the national average values for the US concrete industry (SI Section 7)[60]. $\varphi_M$ for the four transportation modes are obtained from the Ecoinvent database (SI Section 7).

Process 7—Monoethanolamine (MEA) Production: The $CO_2$ impact of carbon capture is the product of the mass of $CO_2$ which is captured and used in the curing or mixing of CCU concrete ($\varphi_{CCU}$, kg $CO_2$) and the life cycle $CO_2$ emissions from producing a monoethanolamine (MEA) post-combustion $CO_2$ capture system ($\varphi_{MEA}$). $\varphi_{MEA}$ is obtained from the literature review of 21 studies[44,64–83] (SI Section 3).

MEA systems are considered as they capture $CO_2$ with a high efficiency (90%)[64,65,84], capture $CO_2$ from dilute concentrations[85], are retrofittable to power plants currently in operation and are a commercially mature technology[86,87]. The power sector accounts for 28% of the overall $CO_2$ emissions in the U.S[88] and is, therefore, a good candidate for carbon capture. As a result, we consider $CO_2$ capture from power plants. Post-combustion capture is considered as it more commonly deployed than oxy-fuel and pre-combustion systems[65,85]. The reader can refer to[65,85] for further details on the underlying physical principles of carbon capture using MEA, which is beyond the scope of this work.

Process 8—Power plant electricity generation: When CCU concrete is produced, the total $CO_2$ emissions from the power plant is the sum of two components.

$$\left(Alloc_{elec} * \varphi_{Not\ Cap} + \varphi_{Avg} * E_p\right)$$

$Alloc_{elec}$ quantifies the allocation of $CO_2$ emissions from a coal power plant between the co-products of electricity and fly ash, which is used as SCM in concrete production in certain datasets. $Alloc_{elec}$ is 0.98 or 0.94 as economic or mass allocation allocates 0.02 and 0.06 of the total $CO_2$ emissions from the coal power plant to the co-product of fly ash (SI Sections 5 and 6). $Alloc_{elec}$ is 1 when electricity is sourced from a combined cycle natural gas power plant or when system boundary expansion is used (instead of economic or mass allocation). $\varphi_{Not\ Cap}$ accounts for the 10% of $CO_2$ which is not captured as the capture efficiency of the MEA system is 90%[64,65,84].

The second component accounts for the $CO_2$ emissions from compensating for the energy penalty ($E_p$ in kWh), which is incurred when $CO_2$ is captured from a power plant. The second component is the product of $E_p$ and the $CO_2$ intensity of the electricity used to compensate for $E_p$ ($\varphi_{Avg}$ in kg $CO_2$/kWh).

$E_p$ is quantified as follows

$$E_p = \varphi_{CCU} * \left[(heat_{ccu} * hte * 0.277) + E_{pump} + E_{liq}\right] \tag{4}$$

$\varphi_{CCU}$ is the mass of $CO_2$, which is captured from a power plant and utilized in CCU concrete production. $heat_{ccu}$ represents the heat required to regenerate the MEA (2.7 to 3.3 MJ/kg $CO_2$, Supplementary Table 5), which could have alternately been used to generate electricity in the power plant[70,89–91]. hte is the heat to electricity factor (0.09 to 0.25, Supplementary Table 5), which is used to determine the electricity equivalent of $heat_{ccu}$. $E_{pump}$ is the electricity required to power the pumps and fans in the carbon capture unit (16.6 to $30.6 \times 10^{-3}$ kWh/kg $CO_2$, Supplementary Table 5) and $E_{liq}$ is the electricity required to liquify the captured $CO_2$ (0.089 kWh/kg $CO_2$, SI Section 3 "$CO_2$ Liquefaction")).

This analysis follows the standards recommended by the National Energy Technology Laboratory (NETL)[92] to determine the $CO_2$ intensity of the electricity used to compensate for the energy penalty. NETL recommends that the energy penalty is compensated through an external electricity source, which is representative of the grid-mix of the region in which the analysis is carried out[92]. $\varphi_{Avg}$ varies between 0.38 and to 0.56 kg $CO_2$/kWh, which represents the lower and upper limit of the average $CO_2$ intensity of electricity generated in the different grid regions in the US in 2020[92].

Process 9—$CO_2$ Transportation: This analysis assumes that the captured $CO_2$ is transported in a semi-trailer truck (SI Section 3 "$CO_2$ Transportation") as it is necessary to supply the $CO_2$ from the site of capture to geographically disperse concrete curing or mixing facilities, which are primarily accessible by road[21]. The $CO_2$ emissions from transporting $CO_2$ is the product of the total weight ($\varphi_{CCU}$ plus the tare weight), the distance over which the transport occurs ($D_{CO2}$ in km) and the $CO_2$ intensity of transportation emissions of a semi-trailer truck ($\varphi_T = 112$ g $CO_2$ per ton km, Supplementary Table 11). The transport of 1 kg of $CO_2$ necessitates the transport of an additional tare weight ($T_w$) of 0.4 kg in the onward trip to the CCU concrete production facility (Supplementary Table 7). In the return trip, we account for the $CO_2$ emissions from the transport of only the tare weight. As a result, $T_w$ equals 0.8. We assume $D_{CO2}$ to be 810 km, which is equal to the longest distance by which $CO_2$ can be transported in the U.S[93].

Processes 10 and 11—Vaporization and injection of $CO_2$: After transportation, the liquified $CO_2$ needs to be vaporized to a gaseous state and injected into the concrete sample for curing or mixing[94]. The $CO_2$ emissions from vaporizing ($\varphi_{Vap}$) and injecting $CO_2$ ($\varphi_{Inj}$) is the product of $\varphi_{CCU}$ (kg $CO_2$), $\varphi_{Avg}$ (kg $CO_2$/kWh) and the electricity required to vaporize ($5.3 \times 10^{-3}$ kWh/kg $CO_2$, SI Section 3) and inject $CO_2$ ($37 \times 10^{-3}$ kWh/kg $CO_2$)[16], respectively. $\eta$ is the $CO_2$ absorption efficiency and represents the portion of the total $CO_2$ which is absorbed during mixing or curing of concrete (datasets 71 to 99). $\eta$ varies between 50% and 85% during mixing[16,19,52]. For curing, $\eta$ is equal to 1 (i.e.,100% absorption) as the

curing datasets (datasets 1 to 70) report $CO_2$ utilized as the ratio of the mass of $CO_2$ absorbed to the mass of cement.

Processes 12 and 13—$CO_2$ and steam curing: The $CO_2$ emissions from $CO_2$ curing of the concrete sample ($\varphi_{CO2\_Cur}$) is the product of $\varphi_{CCU}$ (kg $CO_2$), $\varphi_{Avg}$ (kg $CO_2$/kWh), the electrical power requirements of the curing chamber ($P_{CO2\_Cur} = 38.8$ kW/m$^3$ of concrete)[35,95] and the duration of curing ($t_{CO2\_Cur}$ in hours, SI Section 2), which is determined from the literature review[38,96]. $\varphi_{CO2\_Cur}$ is normalized to the compressive strength of the concrete sample. In some datasets, a combination of steam and $CO_2$ curing is used for the production of CCU concrete. In this case, the analysis includes the $CO_2$ emissions from steam curing of CCU concrete. The $CO_2$ emissions from steam curing ($\varphi_{Stm\_Cur}$) is the product of $CO_2$ intensity of steam curing (39.55 kg $CO_2$/m$^3$/h, Supplementary Table 8) and the duration of steam curing ($t_{stm\_Cur}$ in hours), which is determined from the literature (Supplementary Table 1 Process 13). $\varphi_{Stm\_Cur}$ is normalized to the compressive strength of the concrete sample.

When $CO_2$ is used for mixing of concrete (datasets in category 3 and 4), the $CO_2$ emissions from $CO_2$ and steam curing are assumed to be zero as $CO_2$ curing of concrete is not conducted.

**Conventional concrete production $CO_2$ emissions.** The total life cycle $CO_2$ emissions from producing conventional concrete ($TOT_{Conv}$) are similarly quantified in Eq. 5.

$$TOT_{Conv} = (\varphi_C * C_{Conv}) + (\varphi_{CA} * CA_{conv}) + (\varphi_{FA} * FA_{conv}) + (\varphi_W * W_{conv})$$
$$+ (\varphi_{SCM} * SCM_{conv}) + (E_p * \varphi_{Pow\_Plnt} * Alloc_{elec}) + \varphi_{Stm\_Cur} + (D_M * \varphi_{TM} * M_{Conv}) \quad (5)$$

($E_p * \varphi_{Pow\_Plnt} * Alloc_{elec}$) quantifies the $CO_2$ emissions from generating $E_p$ kWh of electricity in a power plant without carbon capture. $\varphi_{Pow\_Plnt}$ is the $CO_2$ intensity of electricity generated in a coal or NGCC plant (kg $CO_2$/kWh, SI Supplementary Table 1).

**Net $CO_2$ benefit and sensitivity analysis.** The difference between the $TOT_{CCU}$ (Eq. 1) and $TOT_{Conv}$ (Eq. 5) determines the net $CO_2$ benefit from CCU concrete substituting conventional concrete.

$$Net\ CO_2\ Benefit = TOT_{Conv} - TOT_{CCU} \quad (6)$$

$TOT_{CCU}$ and $TOT_{Conv}$ are driven by the $CO_2$ emissions from the 13 processes, which, in turn, are impacted by the uncertainty and variability in the underlying parameters (Supplementary Table 1).

In the scatter plot analysis, 10,000 values are stochastically generated for the material and inventory items and the parameters for the 13 processes, which are obtained from the dataset (ranges and relationships presented in Supplementary Table 1). The stochastically generated values are applied in Eqs. 1, 5, and 6 to determine the $CO_2$ emissions from the 13 processes for conventional and CCU concrete and the net $CO_2$ benefit. The net $CO_2$ benefit is plotted on the ordinate. The difference between the $CO_2$ emissions for each of the 13 contributing processes in conventional and concrete is plotted on the abscissa.

To further verify the results, this analysis conducts a moment independent sensitivity analysis[25,29,30,97] to determine the process (from the 13 processes) having the most impact on the net $CO_2$ benefit. The moment independent sensitivity analysis determines the δ index for each of the 13 processes. The δ index quantifies the relative contribution of each of the 13 processes to the probability distribution function of the net $CO_2$ benefit. The moment independent sensitivity analysis offers methodological advantages as it accounts for the correlation between the input parameters for the 13 processes and is applicable when the input parameters and the output are not linearly related[98]. This study determines the δ indices over 10,000 Monte Carlo runs based on the approach presented in Wei, Lu, and Yuan[97].

## Data availability
All the datasets utilized and analyzed in this study are included in section 2 of the Supplementary information file.

## Code availability
The code used in the analysis is included in section 8 of the Supplementary information file. The code can be accessed on ref. [99].

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

## Acknowledgements

The authors thank Helaine Hunscher and Dr. Christophe Mangin from the University of Michigan for feedback, which improved the quality of the manuscript. This work was supported by the Global $CO_2$ Initiative, Center for Sustainable Systems (CSS), School for Environment and Sustainability (SEAS) and the Blue Sky Program of the College of Engineering at the University of Michigan. D.R. conducted and completed the research presented in this manuscript at the Center for Sustainable Systems, University of Michigan, USA. D.R. is currently affiliated with the National Renewable Energy Laboratory (NREL), USA.

## Author contributions

D.R., G.K., and S.M. designed the research. D.R. reviewed the literature, collated the datasets, conducted the analysis and wrote the Python code. D.R. and D.Z. wrote the first draft of the manuscript. G.K., S.M., V.S., D.Z., and V.L. contributed to improving the analysis and visuals and revisions of the manuscript.

## Competing interests

The authors declare no competing interests.
