## [Peer Review File · Nature Communications]

REVIEWER COMMENTS

Reviewer #1 (Remarks to the Author):

This is a very interesting article in a topic that is attracting significant attention. The message of the article is clear and the authors make an excellent job at illustrating the complexity of the problem. As the devil is always in the details, I have a number of questions regarding the article and the assumptions made in the SI which are not always clear to me.

- In line 44, 45 the authors refer to ref 16 which looks into the impact of humidity in CO₂ uptake in light weight concrete. I am not convinced the conclusion of the reference can then be generalized in the way the authors do it here.
- The use of the functional unit (kg CO₂/Mpa/m³ +Ep Kwh) is ok (as they are using system expansion). The second part of the f.u. however is not used in any of the results, so in practice the results are only expressed per Kg CO₂/Mpa/M³ which is then confusing. Please clarify.
- From figure 1.1. is not clear to me whether upstream emissions from nat gas/coal extraction are included in the picture. Also the figure gives the impression that all of the CO₂ from a coal plant will go into concrete. Given the large amount of CO₂ produced per plant I wonder whether a single concrete production location could handle this. I understand that values are normalized to 1 tonne of CO₂ but I would argue than when designing the system, the system itself should be realistic and if not all CO₂ is likely to be processed in a plant, then part of the CO₂ should be allocated to the plant and the other part to other use (or to storage). That would change the whole system. I get that this would overcomplicate the message but at least a discussion on this should be given in the paper
- The figure 1.1 assumes the uptake of CO₂ in the concrete is 100%. Is this realistic?
- There are several studies analyzing the use of flue gas directly in concrete production. Here it is chosen for using a purified stream, why?
- The red color in figures 2 to 5 makes difficult to read some of the individual processes (e.g., power plant electricity production)
- Line 131, I think the authors mean cement production not cement use
- Line 145 (but also other parts in the paper). The issue of impact of compressive strength is not yet defined. One can find papers reporting increases in strength. It would be good to keep the disagreement in this aspect throughout the argumentation in the text
- Lines 165-170 repetitive argumentation
- Lines 174-177: the information that is not provided (explicitly) in the paper and would be beneficial for the read is the amount of CO₂ that is utilized per tonne of concrete (compare to how much Cement or other materials). Because the main raw material of concrete is cement, it is not surprising that the life cycle CO₂eq of Concrete are mostly determined by the life CO₂eq of cement. In this manuscript, it has been assumed that only the coal power plant has applied CCS. The article would benefit of exploring in a sensitivity case what happens when the cement plant also applies CCS and therefore is significantly decarbonized. I would expect then that the relevance of the other parts of the chain will increase and then the source and footprint of the CO₂ used in the concrete will matter.
- Figure 3 and figure 4, would it be possible to link each plot to the specific databased used? right now is only per category but that makes difficult to understand the differences
- Figure 4 is very difficult to read. For instance I would expect power generation with CCS to be in the right/down quadrant but actually is only visible in part of the figures. Is there a reason for this or is only due to the size and color? Why the lower one goes in the figures the less impacts I see? I assume it has to do with the databases but is not clear what is changing...same with figure 5, why I do not "see" a significant impact of OPC production in about half the figures (positively or negatively) and SCM production appears more often as significant in most figures than OPC ? conceptually, this does not make too much sense so I think I am missing something important
- Regarding the allocation, this is a very interesting analysis and is quite remarkable that the choice of allocation methods does not have a large impact which points out to robustness of the results. I have small question here. In the manuscript you allocate parts of the co₂ emissions from a coal power plant to fly ash, therefore the emissions allocated to electricity should decrease. I assume the authors decreased the co₂ emissions of the electricity mix in the analysis to keep consistency across the different parts of the analysis but is not clear whether they did so and by how much? And, if not, why? Furthermore in Line 392 is specified that the ash comes from a coal power plant without CCS but the CO₂ comes from a coal power with CCS? Why?

- Line 262-267. I do not understand how you could make concrete without OPC. Then you only have water, gravel/sand and CO₂, not concrete.
- On the same line and as previously proposed before, a key recommendation to increase the net life cycle CO₂ is to use low carbon cement. This is now missing.
- Line 340-341: the phrase is confusing.
- Line 418: I do not understand that the emission of a coal power plant without CCS is 1.25 kg CO₂ (line 392) and the energy penalty imposed by the capture units is 1 Kg of CO₂? This is 80% of the CO₂ of the power plan without CCS...
- Line 422: the heat of postcombustion does not correspond to current state of the art: 2.8-3.2 MJ/Kg Co₂. Please note that in S5 the range of values reported in the table started in 3480 but the lower limit in the analysis was 3600, why?
- I am puzzled by the decision to use road transport to move the CO₂. If all the CO₂ of a coal power plant (in the order of Mt CO₂ per year) would need to be transported 810 km to a concrete facility, using road transport would not only be non-environmental but also not economically feasible. For that amount of CO₂ and that distance a pipeline would make more sense. It is also not clear from the assumptions whether the emissions of the return trip (empty truck) have been taken into account. Furthermore, it is not any truck but a heavy truck transporting a compressed/refrigerated liquid, they tend to consume more fuel than a normal heavy truck. The number reported corresponds to a normal heavy truck (in ecoinvent)
- Regarding the liquefaction of CO₂, that requires much more than just electricity. A full liquefaction plant will be needed. I do not see the plant in the analysis, including for instance the refrigerant that needs to be used (e.g., NH₃). Furthermore, I found the value of electricity for CO₂ cooling very low, in literature values between 80-106 kWh/ton are reported for the same temperature range.
- Table S6 reporting source 14 is equal to table 10 reporting an ecoinvent factor.
- Please note that in table s7 the CO₂ vaporiser used here was an experimental one which can deal with only 1 tonne per hour. At industrial scale other options are available including water heated vaporizers. Also note that the CO₂ vaporization the authors in 11, 12 used was at atmospheric temperature (not pressure as stated in the table s7)
- S1 process 8: please revise the equation for the fraction of NotCAP

Reviewer #2 (Remarks to the Author):

Review of NCOMMS-20-19590 "Carbon dioxide utilization in concrete production may or may not produce a climate benefit" by Dwarakanath Ravikumar et al.

June, 2020

The paper presents an assessment of various carbon conversion processes related to the concrete industry. The paper presents solid research and contributes to the current literature related to the LCA of carbon conversion. However, in my opinion, the insights generated are not sufficiently novel to justify publication in Nature Communications. In addition, there are some aspects that would strengthen the contribution of this paper that I have summarized below.

Major Issues

- Overall, there is very little discussion of the novelty of this work. My assessment of the novelty is that the paper adds insights in this field through a quantitative review that generates insights including the drivers of variability in emissions and suggestions on how to reduce the impacts of these technology pathways. However, I see this as an incremental addition to the current literature. The abstract states the main insights from the paper as "Our findings demonstrate that the CO₂ benefit from CCU concrete varies widely and is strongly linked to the amount of cement required to compensate losses in compressive strength and the electricity used in the CO₂ curing process." However, many papers have already been published in this area and have reported the potential benefits and sensitivity analysis to show the drivers and opportunities for emissions reductions. The proposed contributions would be augmented if there was more comparison of this work to other

current work in the field. The authors use previous studies as the basis of their analysis but do not compare the insights from those papers with their own to demonstrate that the current analysis is more robust and/or novel. Three examples of work that should be considered and discussed in the paper are:

1. Monkman and MacDonald., 2017, Cleaner Production, Abstract/Table 6
2. Sanna et al., 2014, Chemical Society Review, Table 7
3. Hepburn et al., 2019, Nature, Figure 3

While two of these papers are included in the reference list, they are not compared to explain what insights are being contributed from this paper that are more robust and/or novel.

- I applaud the authors for attempting to treat uncertainty using a more sophisticated technique beyond the standard Monte Carlo commonly used in the LCA literature. However, aggregating the range of inputs to generate generic distributions seems like a big limitation in this study. Common inputs that lack the context of the studies themselves seem problematic. The context of each study should be considered and unique factors considered in the determination of distributions. If not, some support for this simplification should be provided. Also, are 2000 runs in the Monte Carlo Simulation sufficient to generate stable results? This should be discussed and justified as highly uncertain systems typically require 10,000 or more runs to do this.
- The consideration of strength is important. This goes beyond previous LCAs but still does not go far enough to provide insights about the potential use of these materials. There are issues around practicality of application, lifetime that will also impact the viability and potential of these technologies. Lastly, there are issues related not just to risks but also to perception and risk tolerance and differences in different geographies, contexts that are important to note if not deal with in a quantitative way.

Minor Issues

- There are issues with referencing that concerns me in terms of attention to detail throughout the paper. For example, reference 38 is incomplete.
- The title is not clear nor descriptive of the research presented. Is CCU for concrete not worth doing? It seems like the answer is more nuanced than this and should be part of the paper.
- Abstract: "existing estimates do not account for the life cycle CO₂ impact from the capture, transport and utilization of CO₂". I don't agree with this statement. There are many LCA studies on this topic that have been published that take these stages into account.
- No mention of methods in the abstract related to how CO₂ is being incorporated.
- Line 33: What unit basis is this comparison occurring and what are the other two products? What type of mineralization process is discussed here?
- Line 38: Did your findings support this? As written, it sounds like you are endorsing this estimate while your analysis seems to conclude something different.
- Line 55: There is only benefit to representing the uncertainty and variability accurately. It is possible to do more harm than good if the input values used to inform the distributions are based on guesses or a combination of datasets without appropriately combining these datasets.
- Line 59: how the accounting for uncertainty helps identify hotspots and increase the CO₂ benefits should be made more explicit.
- Line 64: What are the 13 upstream processes?
- Line 65: How did you ensure that the stages were consistent in the boundary? Please describe separately in methods section.
- Line 74: Why are you describing results in the introduction section?
- Line 100: Sentence structure, "...which signifies that the in the 2000....".
- Line 144: How much has the strength decreased by? A percentage would be very helpful in interpreting the impacts.
- Line 146: How much OPC is added/substituted to compensate?
- Line 147: What is the final strength of the CCU and conventional concretes?
- Line 149: How much CO₂ is captured and used in curing vs. what is increased from OPC addition?
- Line 171: What is the source/type of energy?
- Line 223: Why is allocation used? Discuss the rationale.
- Line 289: What is the application and end-product of your study? What does your study about what applications should be recommended moving forward?
- Line 296: Keep tense consistent throughout the paper.
- Line 298: How were these studies located and selected? Is this comprehensive?

- Line 307: What is the compressive strength you selected? How? And does it differ for each of the 99 datasets?
- Line 338-340: Sentence not complete.
- Line 352: Delete one "in"
- Line 437: Awkward sentence: What is the difference between tanker and heavy-duty truck?
- Line 441: "and the CO2" font is different.
- Line 441/442: change to "...the CO2 intensity of transportation emissions of a heavy-duty truck."

Reviewer #3 (Remarks to the Author):

General comments:

This paper presents an analysis for identifying the cost-benefit analysis of CO₂ sequestration in concrete production. Considerable research and development are currently ongoing to investigate the possible use of CO₂ products in concrete production. The work presented is, therefore, relevant and of interest to the readers. The article is also well-written. ***The reviewer suggests that the paper might be accepted for publication pending addressing the reviewer's comments provided below.***

1. The authors need to provide a range of concrete mix variables and concrete characteristics used in the study. For instance, it is crucial to provide a table explaining the range of cement content kg/m³, SCM/cement ratio and SCM kg/m³, water/cement ratio, and concrete compressive strength range and concrete densities considered in the study. The authors might then make their discussion while pointing out that the analysis they conducted is limited to this range of variables and characteristics. It is essential to indicate that this conclusion is not general and might be affected by changes in these variables. This addition will also provide useful information to other researchers to examine which of those variables (in concrete mix) or characteristics (e.g., strength as indicated by the authors) might dictate the net benefit of using CO₂ in concrete.
2. It is also critical to point out the limited technologies considered to include CO₂ in concrete. The data presented by the authors are limited to two techniques where CO₂ was used in fresh concrete or concrete was cured using CO₂. Alternative methods for incorporating CO₂ in concrete have been reported in the literature and used in the industry (e.g., dispersing CO₂ particles in fresh concrete). While it is not expected that the authors will cover all those methods, the authors need to acknowledge this limitation in their introduction and when discussing their results. The conclusion made is strongly related to the limited techniques used for incorporating CO₂ in concrete.
3. There is a definite climate benefit on sequestering CO₂ from the climate into products; the question the authors are answering is whether the net benefit is positive or negative. The discussion and conclusion sections, as well as the title, shall point out to the fact the paper is discussing "net benefit" not "benefit." For instance, in line 20 in the abstract, "benefit" shall be changed to "net benefit." This issue is overlooked in multiple locations throughout the paper and shall be fixed.
4. Given the above comments, it is the reviewer's opinion that the title is overstated and might be a bit misleading. The title indicates that the work covers all types of concretes and all methods for incorporating CO₂ in concrete, and this is not true. The reviewer suggests that the authors shall change the title to indicate this is a preliminary or limited set analysis and that the benefit considered herein is the net benefit something like "*Preliminary analysis indicates carbon utilization in concrete production may or may not produce a net climate benefit*".

REVIEWER COMMENTS

Reviewer #1 (Remarks to the Author):

This is a very interesting article in a topic that is attracting significant attention. The message of the article is clear and the authors make an excellent job at illustrating the complexity of the problem. As the devil is always in the details, I have a number of questions regarding the article and the assumptions made in the SI which are not always clear to me.

- In line 44, 45 the authors refer to ref 16 which looks into the impact of humidity in CO₂ uptake in light weight concrete. I am not convinced the conclusion of the reference can then be generalized in the way the authors do it here.

Response: We addressed this review comment by conducting a literature review to identify 31 datasets wherein the compressive strength of CCU concrete is lower than conventional concrete. We have modified text in the main manuscript (lines 57-60) and included new analysis in the SI (section 2).

Lines 57-60

“CO₂ curing can decrease the compressive strength of CCU concrete when compared to conventional concrete. For example, a review of 99 experimental datasets from existing literature shows that CCU concrete has a lower compressive strength than conventional concrete in 31 datasets (SI Section 2 Figure S3).”

• The use of the functional unit (kg CO₂/Mpa/m³ +E_p Kwh) is ok (as they are using system expansion). The second part of the f.u. however is not used in any of the results, so in practice the results are only expressed per Kg CO₂/Mpa/M³ which is then confusing. Please clarify.

Response: We have modified the text to improve the articulation on how the CO₂ impact of E_p is accounted for in equation 1 and equation 4.

Equation 1, which quantifies the total CO₂ emissions from CCU concrete production, includes an expression to calculate the CO₂ impact of E_p

Equation 1 - Lines 429

$(\varphi_{Avg} * E_p)$

- φ_{Avg} is the average CO₂ intensity of the external electricity source used to compensate for E_p
- E_p is the energy penalty per kg of CO₂ captured

Lines 496- 499

“The second component accounts for the CO₂ emissions from compensating for the energy penalty (E_p in kWh), which is incurred when CO₂ is captured from a power plant. The second component is the product of E_p and the CO₂ intensity of the electricity used to compensate for E_p (φ_{Avg} in kg CO₂/kWh).”

Furthermore, Equation 4 presents the expression used to determine E_p.

• From figure 1.1. is not clear to me whether upstream emissions from nat gas/coal extraction are included in the picture.

Response: We have included a text box stating “Coal/NGCC Extraction” in the top left corner of figure 1 in the revised manuscript.

Also the figure gives the impression that all of the CO₂ from a coal plant will go into concrete. Given the large amount of CO₂ produced per plant I wonder whether a single concrete production location could handle this. I understand that values are normalized to 1 tonne of CO₂ but I would argue than when designing the system, the system itself should be realistic and if not all CO₂ is likely to be processed in a plant, then part of the CO₂ should be allocated to the plant and the other part to other use (or to storage). That would change the whole system. I get that this would overcomplicate the message but at least a discussion on this should be given in the paper

Response: We have modified the figure and text in the caption and the manuscript to clearly communicate that the CO₂ captured at the power plant is equal to the mass of CO₂ required for the production of CCU concrete (Φ_{CCU} in kg). The CO₂ required for the CCU concrete is obtained from the dataset, which is presented in section 2 of the SI.

To facilitate clearer understanding, we have indicated the amount of CO₂ captured by the “ Φ_{CCU} ” term in Figure 1 with an explanation in the caption.

Lines 98-107

“Figure 1 The processes required to produce CCU concrete are highlighted in grey and green. The processes required to manufacture conventional concrete are highlighted in grey and red. The CO₂ emissions, which is utilized in the curing or mixing of CCU concrete (ϕ_{CCU} in kg), is captured from a power plant. The energy penalty from capturing ϕ_{CCU} (E_p kWh) is compensated by an external power plant. When conventional concrete is produced, there is no carbon capture during electricity generation and the CO₂ from generating E_p in the power plant is completely emitted. The functional unit - 1 m³ of concrete with 1 MPa strength and E_p kWh of electricity - is common across the CCU and conventional concrete production pathways. The CO₂ emissions from each of the CCU and conventional concrete production process is quantified in Equations 1 and 5.”

We have included text to account for the limitation identified by the reviewer

Lines 326-335

“The system boundary (Figure 1) assumes that the CO₂ captured from the power plant is used for CCU concrete production without any intermediate storage. In practice, the total CO₂ captured from a power plant may be significantly greater than the maximum utilization capacity at a CCU concrete production plant. In such cases, the excess captured CO₂ may be temporarily stored for future utilization in CCU concrete production or routed towards other utilization

pathways. Given that CO₂ utilization is an emerging field and in the early stages of commercialization, there is a lack of time-sensitive data on how the captured CO₂ feedstock is either temporarily stored or immediately allocated to other utilization pathways. As a result, a system boundary that incorporates the time-sensitive utilization of CO₂ captured from a power plant is beyond the scope of this work and is a topic for future research.”

- The figure 1.1 assumes the uptake of CO₂ in the concrete is 100%. Is this realistic?

Response: Based on value reported in literature, we have assumed that only 50 to 85% of the captured CO₂ is used during mixing. For curing, 100% of the captured CO₂ is used. We have accounted for this through the following expression in equation 1.

$$(\varphi_{CCU} * (\varphi_{Inj} + (1 - \eta)))$$

We have included text in the main manuscript and in the SI to explain this.

Lines 531-535

η is the CO₂ absorption efficiency and represents the portion of the total CO₂ which is absorbed during mixing or curing of concrete (datasets 71 to 99). η varies between 50% to 85% during mixing.^{16,19,52} For curing, η is equal to 1 (i.e.100% absorption) as the curing datasets (datasets 1 to 70) report CO₂ utilized as the ratio of the mass of CO₂ absorbed to the mass of cement.”

SI

Please refer process 11 in SI Table 1.

- There are several studies analyzing the use of flue gas directly in concrete production. Here it is chosen for using a purified stream, why?

Response: We choose pure CO₂ in this analysis as it is the most commonly reported type of CO₂ feedstock with the most amount of experimental data available in studies comparing the CCU and conventional concrete.

In the revised manuscript, we have included new text on how there are alternatives such as the direct use of flue gas, which can be used instead of purified CO₂. The framework of analysis presented in this study can be extended in the future when data

comparing the material properties of CCU concrete (from flue gas) is more widely available.

Lines 344-355

“This analysis focusses on the use of pure CO₂ and two approaches of CO₂ utilization - curing and mixing - as they are more extensively investigated (e.g. 99 datasets used in this study) than alternate approaches such as concrete curing with flue gas,^{21,45-47} carbonation of recycled concrete aggregates,⁹ CO₂ sequestration in alternative MgO based binders,¹⁰ and CO₂ dissolution in mixing water.^{13,14} The increased availability of experimental data is necessary to robustly quantify the net CO₂ benefit of CCU concrete and account for the impact of data uncertainty and process variability on the results. For example, further experimental research can generate data on the variability in the compressive strength of flue-gas cured concrete properties when the 2-week curing time is reduced⁴⁷ and the CO₂ concentration in the flue gas is varied.⁴⁵ With increased availability of inventory and process data from future experimental research, the life cycle approach presented in this study can be extended to quantify the net CO₂ benefit of CCU concrete produced from flue gas and other alternate approaches of CO₂ utilization.”

- The red color in figures 2 to 5 makes difficult to read some of the individual processes (e.g., power plant electricity production)

Response: We have changed the colors used so that all of the individual processes are visible when the green and red background colors are used in Figures 2 to 5.

- Line 131, I think the authors mean cement production not cement use

Response: We have changed the text to cement production.

Lines 142-144

“Therefore, for dataset 1, the difference between the CO₂ emissions from cement production is the most important reason for TOT_{CCU} being greater than TOT_{Conv} (i.e. the net CO₂ benefit being negative).”

• Line 145 (but also other parts in the paper). The issue of impact of compressive strength is not yet defined. One can find papers reporting increases in strength. It would be good to keep the disagreement in this aspect throughout the argumentation in the text

Response: We have modified the text to mention that the finding is specific dataset 1 used in the illustrative example and included the values specific to this dataset to help the reader better understand the results

Lines 155-161

“The finding from the sensitivity analysis for dataset 1 can be attributed to the compressive strength of CCU concrete (16-17.4 MPa) being lower than conventional concrete (18 – 18.6 MPa). The mean compressive strength of CCU concrete (16.7 MPa) is 9% lower than that of conventional concrete (18.3 MPa). This implies that in dataset 1 a greater mass of OPC is produced for CCU concrete to achieve the same compressive strength as conventional concrete. In dataset 1, the OPC produced per MPa for CCU concrete is 24.8 kg/MPa and for conventional concrete is 22.6 kg/MPa.”

Moreover, we have included text in the introduction section to explain that CO₂ curing decreases the compressive strength of CCU concrete when compared to conventionally cured concrete in 31 of the 99 datasets.

Lines 57 to 64

“CO₂ curing can decrease the compressive strength of CCU concrete when compared to conventional concrete. For example, a review of 99 experimental datasets from existing literature shows that CCU concrete has a lower compressive strength than conventional concrete in 31 datasets (SI Section 2 Figure S3). In such cases, CCU concrete would require a greater amount of OPC than conventional concrete to produce the same compressive strength.

OPC production is a major source of CO₂ emissions. Therefore, increased OPC content in a concrete formulation leads to an increase in CO₂ emissions from upstream cement production processes, which may outweigh the benefit of the CO₂ captured and used in concrete production.”

- Lines 165-170 repetitive argumentation

Response: We have removed the repetitive text and made this section briefer from the original 97 words to the 67 words in the revised manuscript.

Lines 184 to 189

“The compressive strength in CCU concrete decreases due to CO₂ curing when compared to conventionally cured concrete. The OPC and SCM consumed to produce the same compressive strength is greater in CCU concrete than in conventional concrete. Therefore, the results demonstrate the CO₂ burden of increased OPC and SCM consumption for CCU concrete outweighs the benefit of the CO₂ that is captured and used in CCU curing.”

- Lines 174-177: the information that is not provided (explicitly) in the paper and would be beneficial for the read is the amount of CO₂ that is utilized per tonne of concrete (compare to how much Cement or other materials). Because the main raw material of concrete is cement, it is not surprising that the life cycle CO₂eq of Concrete are mostly determined by the life CO₂eq of cement.

Response: We have provided the amount of CO₂, cement, coarse aggregate, fine aggregate and water used per m³ of the concrete in the excel sheet linked in SI section 2 (available here: <https://www.dropbox.com/s/ueex0uk3m95q3eo/Literature.xlsx?dl=0>).

We have now extracted and presented the data visually in a Figure S1 to Figure S5 in the SI Section 2 so that the reader can examine the design mix and material used to produce CCU concrete.

We have included text in the main manuscript to clarify what the review comment has mentioned

Lines 195-197

“This can be attributed to the mass of CO₂ utilized in concrete being lower than the mass of the cement and coarse and fine aggregate (SI Figure S1) and the life cycle CO₂ intensity of coarse and fine aggregate being significantly lower than cement (SI Table S2).”

Furthermore, in the methods section, we refer the reader to section 2 in the SI to understand the materials used in the design mix of CCU concrete.

Lines 372-374

(i) the design mix consisting of the energy and material inventory required for the production of conventional and CCU concrete (SI Section 2).”

In this manuscript, it has been assumed that only the coal power plant has applied CCS. The article would benefit of exploring in a sensitivity case what happens when the cement plant also applies CCS and therefore is significantly decarbonized. I would expect then that the relevance of the other parts of the chain will increase and then the source and footprint of the CO₂ used in the concrete will matter.

Response: We have now expanded our analysis and included a new scenario where CO₂ is captured from a cement plant and utilized in concrete production. The detailed results are included in Section 11 of the SI.

As the reviewer mentions, the influence of the OPC production decreases. Datasets in which CCU had a higher CO₂ impact than conventional concrete due to increased cement use (per MPa basis) are now favorable for CCU as CO₂ is captured in the CCU cement plant.

We have included text in the main manuscript to refer to the results from the scenario analysis.

Lines 198 to 206

“To investigate the change in results when CO₂ intensity of OPC production decreases, we determine the net CO₂ benefit of CCU concrete when CO₂ is captured from a cement plant (SI

Section 11). The results show that CCU concrete has higher life cycle CO₂ emissions than conventional concrete (i.e. negative CO₂ benefit) in 44 out of the 99 datasets (Figure 4 in SI Section 11) when compared to 56 out of the 99 datasets in the baseline scenario (Figure 5). Therefore, when CO₂ is captured from a cement plant, there is a lower likelihood of CCU concrete producing a negative net CO₂ benefit than when CO₂ is captured from a power plant. The difference in the results can be attributed to the reduced CO₂ intensity of cement production due to CO₂ capture at the cement plant.”

• Figure 3 and figure 4, would it be possible to link each plot to the specific databased used? right now is only per category but that makes difficult to understand the differences

Response: The dataset numbers are included in parenthesis on the left at the bottom of each graph.

The dataset number and material and energy inventory details for each of the datasets is provided in section S2 of the SI.

• Figure 4 is very difficult to read. For instance I would expect power generation with CCS to be in the right/down quadrant but actually is only visible in part of the figures. Is there a reason for this or is only due to the size and color? Why the lower one goes in the figures the less impacts I see? I assume it has to do with the databases but is not clear what is changing...same with figure 5, why I do not “see” a significant impact of OPC production in about half the figures (positively or negatively) and SCM production appears more often as significant in most figures than OPC ? conceptually, this does not make too much sense so I think I am missing something important

Response: In the scatter plot, the limits of the x-axis are determined by the process with the greatest CO₂ emissions. For example, in plot 10 in Figure 4 in the main paper, the lower and the upper limit of the x-axis is -10 to +10, which is driven by the CO₂ emissions from OPC production (purple scattered points for process 1). With these limits of the x-axis, the scattered points corresponding to processes with significantly lower CO₂ emissions cannot be viewed on the plot. For example, the maximum and minimum values for the x-axis for process 2 (coarse aggregate production) is -0.05 to +0.05 and, as a result, cannot be viewed on the x-axis ranging between -10 and +10. We have explained this in SI Section 9.

To allow the reader to view the scatter points for each process, we have generated the scatterplots individually for the 13 parameters, which can be downloaded from Table S12 in the SI. The reader can download and expand and view the images of the scatter plot for each of the 13 processes at a higher resolution.

Figure 4 shows that there are three key processes with significant CO₂ emissions that contribute the most to the Net CO₂ benefit – P1:OPC production, P5: SCM production and P12: CO₂ curing. The lower we move down in the plots, the x-axis limits are dominated by the range of the key processes of OPC and SCM production, which decreases the resolution for other processes. At this decreased resolution, we can only see the scattered points for OPC and SCM production.

As indicated by the results in the sensitivity analysis (Figure 5), SCM is a key driver in certain datasets (51-70, 79-99). This is a function of the increased mass of SCM consumed and the CO₂ footprint of SCM production, which is byproduct of the coal electricity and the pig iron production and determined by the system boundary expansion, mass allocation and economic allocation.

We have included text in the manuscript to explain that OPC production, SCM production and energy used for CO₂ curing are the 3 key processes driving the net CO₂ benefit of CCU concrete.

Lines 177-191

“A visual inspection of the slopes in the scatter plot in Figure 4 and the delta indices in Figure 5 reveal that the net CO₂ benefit is most sensitive to the amount of OPC produced and used in the design mix (e.g. P1 in datasets 1, 2, 3 and 4 in Figure 5), the energy used for CO₂ curing (e.g. P12 in datasets 21, 22 and 23 in Figure 5) and SCM produced and used (e.g. P5 in datasets 51, 52, 53 and 54 in Figure 5).

The plots with the red background in Figure 3, Figure 4 and Figure 5 demonstrate that the net CO₂ benefit is negative (i.e. CCU concrete has higher life cycle CO₂ emissions than conventional concrete) with at least a 50% likelihood in 56 out of the 99 datasets. The compressive strength in CCU concrete decreases due to CO₂ curing when compared to conventionally cured concrete.

The OPC and SCM consumed to produce the same compressive strength is greater in CCU concrete than in conventional concrete. Therefore, the results demonstrate the CO₂ burden of increased OPC and SCM consumption for CCU concrete outweighs the benefit of the CO₂ that is captured and used in CCU curing. Additionally, in category 1 datasets, the electricity use in the

CO₂ curing process is the second key contributor to the increase in the total life cycle CO₂ emissions from CCU concrete (e.g. datasets 46, 47 and 48 in Figure 5)."

• Regarding the allocation, this is a very interesting analysis and is quite remarkable that the choice of allocation methods does not have a large impact which points out to robustness of the results. I have small question here. In the manuscript you allocate parts of the co2 emissions from a coal power plant to fly ash, therefore the emissions allocated to electricity should decrease. I assume the authors decreased the co2 emissions of the electricity mix in the analysis to keep consistency across the different parts of the analysis but is not clear whether they did so and by how much? And, if not, why?

Response: We have modified text to explain the allocation between fly ash and electricity from a coal plant.

The allocation is accounted for through the 'Alloc_{ash}' and 'Alloc_{elec}' terms in process 5 (in Equations 3) and process 8 when CCU and conventional concrete is produced

Allocating CO₂ emissions to Fly Ash for CCU Concrete

Lines 458-463

Process 5 – SCM production

"When fly ash is used as the SCM, the CO₂ impact per kg of fly ash (φ_{SCM_ash} in kg CO₂/kg fly ash) is determined from Equation 3

$$\varphi_{SCM_ash} = Alloc_{ash} * 22.7 * \varphi_{Elec_Coal} * \alpha_{Cap} \quad \text{Equation 3}$$

The value of Alloc_{ash} is 1, 0.02 or 0.06 when SE, MA or EA is chosen, respectively (SI Sections 4, 5 and 6). φ_{Elec_Coal} , which is the life cycle CO₂ emission from producing 1 kWh of coal electricity, is 1.25 kg CO₂/kWh (SI Section 4)."

Allocating CO₂ emissions to coal electricity for CCU Concrete

Lines 486-494

"Process 8 – Power plant electricity generation: When CCU concrete is produced, the total CO₂ emissions from the power plant is the sum of two components.

$$(Alloc_{elec} * \varphi_{Not\ Cap} + \varphi_{Avg} * E_p)$$

Alloc_{elec} quantifies the allocation of CO₂ emissions from a coal power plant between the co-products of electricity and fly ash, which is used as SCM in concrete production in certain datasets. Alloc_{elec} is 0.98 or 0.94 as economic or mass allocation allocates 0.02 and 0.06 of the total CO₂ emissions from the coal power plant to the co-product of fly ash (SI Sections 5 and 6). Alloc_{elec} is 1 when electricity is sourced from a combined cycle natural gas power plant or when system boundary expansion is used (instead of economic or mass allocation).”

Conventional Concrete

Similarly, for conventional concrete production, the allocation between coal electricity and fly ash has been accounted for in the expressions for Process 5 and Process 8 in Table S1 in the SI Section 1.

Furthermore, in Line 392 is specified that the ash comes from a coal power plant without CCS but the CO₂ comes from a coal power with CCS? Why?

Response: We have modified text and included ‘ α_{Cap} ’ in Equation 3. ‘ α_{Cap} ’ accounts for carbon capture in a coal plant when quantifying the CO₂ emissions from fly ash production.

Lines 458-464

“When fly ash is used as the SCM, the CO₂ impact per kg of fly ash (φ_{SCM_ash} in kg CO₂/kg fly ash) is determined from Equation 3

$$\varphi_{SCM_ash} = Alloc_{ash} * 22.7 * \varphi_{Elec_Coal} * \alpha_{Cap} \quad \text{Equation 3}$$

The value of Alloc_{ash} is 1, 0.02 or 0.06 when SE, MA or EA is chosen, respectively (SI Sections 4, 5 and 6). φ_{Elec_Coal} , which is the life cycle CO₂ emission from producing 1 kWh of coal electricity, is 1.25 kg CO₂/kWh (SI Section 4). α_{Cap} is 0.1 if CO₂ is captured at a coal plant and used in CCU concrete production.”

- Line 262-267. I do not understand how you could make concrete without OPC. Then you only have water, gravel/sand and CO₂, not concrete.

Response: We have modified the text to make it clear that we are referring to how binder material (OPC or SCM) use in CCU concrete can be decreased when compared to conventional concrete through an increase in compressive strength from CO₂ curing.

Lines 289 - 294

“Ensure increase in compressive strength from CO₂ curing: A key priority is to determine a CO₂ curing protocol that consistently increases compressive strength of CCU concrete. An increase in compressive strength implies that a smaller quantity of carbon intensive binder material is used in CCU concrete to achieve the same compressive strength as conventional concrete (i.e. lower quantity of OPC or SCM is consumed on a kg per MPa basis).”

- On the same line and as previously proposed before, a key recommendation to increase the net life cycle CO₂ is to use low carbon cement. This is now missing.

Response: As seen in the response to the previous comment, we have included low carbon cement (SCM) in the discussion. We have modified the text to communicate that an increase in compressive strength from CO₂ curing will result in a smaller mass of binder material being consumed in CCU concrete than conventional concrete to produce the same compressive strength. Binder material includes supplementary cementitious materials, which is a low carbon alternative to cement.

- Line 340-341: the phrase is confusing.

Response: We have modified the text to simplify the phrase and improve readability.

Lines 416-417

“E_p is determined based on the mass of CO₂ captured from the power plant (SI Table SI Process 8).”

- Line 418: I do not understand that the emission of a coal power plant without CCS is 1.25 kg CO₂ (line 392) and the energy penalty imposed by the capture units is 1 Kg of CO₂? This is 80% of the CO₂ of the power plan without CCS...

Response: We have modified the text to address this comment and improve clarity.

Lines 496-499

“The second component accounts for the CO₂ emissions from compensating for the energy penalty (E_p in kWh), which is incurred when CO₂ is captured from a power plant. The second component is the product of E_p and the CO₂ intensity of the electricity used to compensate for E_p (φ_{Avg} in kg CO₂/kWh).”

• Line 422: the heat of postcombustion does not correspond to current state of the art: 2.8-3.2 MJ/Kg Co2. Please note that in S5 the range of values reported in the table started in 3480 but the lower limit in the analysis was 3600, why?

Response: We have modified the heat requirement, which is now 2.7 to 3.3 MJ/kg CO₂.

Lines 503-504

“heat_{ccu} represents the heat required to regenerate the MEA (2.7 to 3.3 MJ/kg CO₂, SI Table S5), which could have alternately been used to generate electricity in the power plant.”

We have taken care to use the same values in the main paper and the SI (Table S5)

• I am puzzled by the decision to use road transport to move the CO₂. If all the CO₂ of a coal power plant (in the order of Mt CO₂ per year) would need to be transported 810 km to a concrete facility, using road transport would not only be non-environmental but also not economically feasible. For that amount of CO₂ and that distance a pipeline would make more sense.

Response: We thank the reviewer for the feedback as it helps better articulate the assumptions and contextualize the findings. We choose road-based transportation as it helps offer the flexibility of transporting the CO₂ to the concrete production facility (as explained in **Lines 516-519**.)

We have also noted the limitation of road-based transportation in the manuscript and explained that pipeline transport has a lower environmental impact (**Lines 336-337**).

Moreover, as indicated by the sensitivity analysis (main paper Figure 5), the transportation of CO₂ is not a significant contributor to the overall net CO₂ impact of CCU concrete (Process P9).

To account for the maximum possible decrease in CO₂ impact of transporting CO₂, we have included a new optimistic scenario in the SI wherein the transport of CO₂ is zero (i.e. CO₂ emissions from process 9 is zero). The results (SI Section 12) are similar to the baseline scenario (main paper Fig 5) where CCU concrete has a lower net CO₂

impact than conventional concrete in 44 datasets. As a result, switching to pipelines to transport CO₂ will not impact the results.

We have included text in the main paper (**Lines 336-343**) and **SI Section 12** to address this review comment.

Lines 336-343

“The transport of CO₂ through pipelines has a lower CO₂ impact than road-based transport using semi-trailer trucks,⁴⁴ which is modeled in this analysis. To quantify the maximum possible gains from shifting to a less carbon intensive mode of CO₂ transport, we conduct a scenario analysis with the optimistic assumption that the CO₂ impact of CO₂ transportation is zero (SI Section 12). Despite this optimistic assumption of zero-carbon CO₂ transport, CCU concrete has a lower CO₂ impact than conventional concrete in 44 of the 99 datasets, which is similar to the 43 of the 99 datasets obtained in the baseline scenario (SI Figure S10 versus Figure 5). As a result, a shift from road to pipeline based CO₂ transport will not impact the findings from this analysis.”

It is also not clear from the assumptions whether the emissions of the return trip (empty truck) have been taken into account. Furthermore, it is not any truck but a heavy truck transporting a compressed/refrigerated liquid, they tend to consume more fuel than a normal heavy truck. The number reported corresponds to a normal heavy truck (in eco-invent)

Response: We have modified the text to improve the explanation on the how the CO₂ emissions of CO₂ transport through road is quantified.

We account for the CO₂ emissions for the onward and return trips and the transport of the tare weight in a semi-trailer truck, which has a total weight of 35 tons (SI Section 3, “CO₂ Transportation” Table S7).

We modeled the CO₂ emissions for transport using a 35-ton semi-trailer ($\varphi_T = 112$ g CO₂ per ton km) based on the eco-invent dataset for a freight truck with a capacity greater than 32 tons.

Please refer SI Section 3, “CO₂ Transportation” Table S7.

- Regarding the liquefaction of CO₂, that requires much more than just electricity. A full

liquefaction plan will be needed. I do not see the plant in the analysis, including for instance the refrigerant that needs to be used (e.g., NH₃). Furthermore, I found the value of electricity for CO₂ cooling very low, in literature values between 80-106 kWh/ton are reported for the same temperature range

Response: We conducted a literature review to determine the liquefaction plan used in our analysis. Based on the literature review, we obtained 22 datapoints for CO₂ liquefaction using ammonia (SI Table S6). We use a value of 89 kWh to liquify 1 tonne of CO₂ to 20 bars based on the liquefaction plan reported in

“Deng, H.; Roussanaly, S.; Skaugen, G., Techno-economic analyses of CO₂ liquefaction: Impact of product pressure and impurities. International Journal of Refrigeration 2019, 103, 301-315.”

In the SI Section 3 subsection “CO₂ Liquefaction”, we refer the reader to the above reference for further details on the liquefaction plan.

Please refer the section titled “CO₂ Liquefaction” in SI Section 3.

- Table S6 reporting source 14 is equal to table 10 reporting an ecoinvent factor.

Response: As reported in Table S11 in the SI, we ensure consistency by obtaining the CO₂ impact of the various modes of transportation from Ecoinvent. We do not use source 14 in the revised manuscript.

- Please note that in table s7 the CO₂ vaporiser used here was an experimental one which can deal with only 1 tonne per hour. At industrial scale other options are available including water heated vaporizers. Also note that the CO₂ vaporization the authors in 11, 12 used was at atmospheric temperature (not pressure as stated in the table s7)

Response: Based on the specifications provided by commercial manufacturers of CO₂ vaporizers, we have tabulated the electricity required for CO₂ liquification in Table S8 of the SI.

We use the lowest value of 5.3 kWh/ton CO₂ for the vaporizer manufactured by Asco.

Please refer the section titled “CO₂ Vaporization” and Table S8 in Section 3 of the SI.

- S1 process 8: please revise the equation for the fraction of NotCAP

Response: From the total CO₂ that is emitted from the power plant, 90% is captured (ϕ_{CCU}) and 10% is not captured ($\phi_{NOT\ Cap}$). As a result, $\phi_{NOT\ Cap} = (1/9) \phi_{CCU}$

We have provided the explanation for process 8 in Table S1.

Reviewer #2

Review of NCOMMS-20-19590 “Carbon dioxide utilization in concrete production may or may not produce a climate benefit” by Dwarakanath Ravikumar et al.

June, 2020

The paper presents an assessment of various carbon conversion processes related to the concrete industry. The paper presents solid research and contributes to the current literature related to the LCA of carbon conversion. However, in my opinion, the insights generated are not sufficiently novel to justify publication in Nature Communications. In addition, there are some aspects that would strengthen the contribution of this paper that I have summarized below.

Response: We thank the reviewer for the constructive feedback, which has helped to improve the articulation of the novelty of the analysis and contrast it with existing work on quantifying the CO₂ impact of CCU concrete.

We have provided further details on how we have accounted for uncertainty and variability in the CCU production conditions based on the inventory data collected from the 99 dataset and the life cycle data from ecoinvent. As a result, we have avoided guesses when quantifying the net CO₂ impact of CCU concrete.

A detailed response to each of the review comments is provided below.

Major Issues

- Overall, there is very little discussion of the novelty of this work. My assessment of the novelty is that the paper adds insights in this field through a quantitative review that generates insights including the drivers of variability in emissions and suggestions on how to reduce the impacts of these technology pathways. However, I see this as an incremental addition to the current literature. The abstract states the main insights from the paper as “Our findings demonstrate that the CO₂ benefit from CCU concrete varies widely and is strongly linked to the amount of cement required to compensate losses in compressive strength and the electricity used in the CO₂ curing process.” However, many papers have already been published in this area and have reported the potential benefits and sensitivity analysis to show the drivers and opportunities for emissions reductions. The proposed contributions would be augmented if there was more comparison of this work to other current work in the field.

Response: We justify the novelty of our manuscript based on two key aspects

1. **Novelty in findings:** Our manuscript is the first to present findings based on a comprehensive life cycle analysis of existing experimental studies and show that CCU concrete produced from CO₂ curing or mixing is more likely to increase in life cycle CO₂ emissions. The findings are in contrast to previous studies, which

claim that CCU concrete decreases CO₂ emissions through CO₂ sequestration. We have modified the title, abstract and the results section to better communicate the novelty of the findings. Examples include,

Title - *“Carbon dioxide utilization in concrete curing or mixing might not produce a net climate benefit”*

Abstract - Lines 18-20

“The results demonstrate a higher likelihood of the net CO₂ benefit of CCU concrete being negative i.e. there is a net increase in CO₂ in 56 to 68 of 99 published experimental datasets depending on the CO₂ source.”

Lines 246-252

“Negative CO₂ net benefit values are obtained in the remaining 56 to 64 datasets. A similar analysis for CO₂ capture from a NGCC power plant shows that the net CO₂ benefit is negative in 61, 65 and 68 of the 99 datasets when SE, MA and EA are used, respectively (SI Section 10). The overall results demonstrate that the CO₂ benefit of CCU concrete production is negative in 56 to 68 of the 99 datasets depending on whether CO₂ is captured from a coal or a NGCC power plant and when SE, MA or EA is used. As a result, there is a higher likelihood of the net CO₂ benefit of CCU concrete being negative.”

- 2. Novelty and robustness in the method used to determine the Net CO₂ benefit of CCU concrete:** To demonstrate how the methods used in this study is different from previous studies, we include a new quantitative literature review of 23 studies in the revised manuscript (SI Section 13). The literature review identifies knowledge gaps and shortcomings in existing literature when determining the net life cycle CO₂ benefit of CCU concrete.

We have referred to this new literature review in the main paper

Lines 48-56

“Estimates show that 0.1 to 1.4 gigatons of CO₂ can be utilized in concrete by 2050.^{1,3} However, a literature review (SI Section 13) demonstrates these estimates are not based on a comprehensive assessment which accounts for the change in compressive strength of concrete from CO₂ utilization; the CO₂ impact of capturing, transporting and utilizing CO₂; the CO₂ emissions from compensating for the energy penalty of CO₂ capture and producing supplementary cementitious materials (SCM), which are by-products of coal electricity and pig iron production; the uncertainty and variability in inventory data and process parameters; and may not always be based on primary experimental data, which is required for a robust life cycle CO₂ assessment.”

Please refer SI Section 13 for the quantitative literature review, which is reproduced below.

“The literature review covered 23 studies containing claims on the CO₂ benefit of CCU concrete. The studies with reference numbers can be found here: <https://rb.gy/afq1te>

The Venn diagram represents the organization of literature review to identify knowledge gaps and shortcomings when determining the CO₂ impact of CCU concrete.

The literature review is organized based on whether a study accounts for the following 5 aspects when quantifying the net CO₂ impact of CCU concrete

- *CO₂ impact of capturing, transporting, compressing and utilizing CO₂: The CO₂ emissions from the upstream processes of capturing, transporting, compressing and utilizing CO₂ will decrease the net CO₂ benefit from CCU concrete. Only 3 (A[1] + E[2]) out of the 23 studies account for the upstream processes. However, none of the 3 studies account for data uncertainty and only 1 study accounts for change in compressive*

strength. None of the 3 studies account for the compensation of the energy penalty of CO₂ capture and allocation of CO₂ emissions to the supplementary cementitious materials (SCM) (SI Section 6), which are by-products of coal electricity and pig-iron production

A: 14, B: 20-23, C: 2, 3, 5-8, 10-13, D: 1, 4, 9, 17-19, E: 15, 16

Figure S2 The 5 ovals in the Venn diagram represent 5 key aspects in a study that are required to determine and robustly substantiate the claims on the CO₂ impact of CCU concrete on a life cycle basis. The numbers within the square parenthesis represent the number of studies in 5 regions of the Venn diagram (A, B, C, D and E). The studies corresponding to the parenthesized numbers for the 5 regions are listed below the Venn diagram. For example, studies 15 and 16 are the 2 studies in region E (i.e. E[2]) of the Venn Diagram.

- *uncertainty and variability in data, allocation for supplementary cementitious materials (SCM) and compensating for energy penalty: As observed in SI Section 2, there is significant variability in the materials used in design mixes across the studies. In addition, there is uncertainty in the life cycle data which will introduce uncertainty in the CO₂ footprint of the material and energy inventory. For example, the uncertainty in the life cycle CO₂ footprint of the material components used in the concrete is presented in Table S1.*

The SCMs - fly ash and slag - are by-products of coal electricity generation and pig-iron production. As a result, when determining the CO₂ footprint of SCMs used in CCU concrete, the analysis should account for the allocation of CO₂ emissions between (i) flash ash and coal electricity and (ii) slag and pig-iron (SI Table S1 Process 5).

The capture of CO₂ (to be utilized in CCU concrete production) from a power plant incurs an energy penalty (SI Table S1 Process 8). The CO₂ emitted from generating electricity from an external power plant to compensate for energy penalty needs to be accounted for when determining the CO₂ impact of CCU concrete.

None of the studies account for data uncertainty, allocation of CO₂ emissions between the by-products SCM and coal electricity and steel, and the energy penalty of CO₂ capture.

- *change in compressive strength when quantifying the climate benefit of CCU concrete: Ordinary Portland Cement (OPC) is the most CO₂ intensive material component in concrete. The increased use of OPC to compensate for loss in compressive strength in CCU concrete can significantly increase the CO₂ footprint of CCU concrete on a life cycle basis.*

As shown in Figure S2, only 1 (A[1]) of the 23 studies accounts for the change in compressive strength. However, this study does not account for data uncertainty, allocation of CO₂ emissions between the by-products SCM and coal electricity and steel, and the energy penalty of CO₂ capture.

- *experimental data comparing compressive strength of conventional and CCU concrete: This helps distinguish between studies that quantify the CO₂ impact of CCU concrete from (i) inventory data collected from experiments comparing the compressive strengths of CCU and conventional concrete and (ii) a review of other studies. The net CO₂ benefit of CCU concrete determined from primary experimental data is more robust as (i) there is clarity and transparency in system boundary assumptions and inventory items consumed (ii) CO₂ hotspots are identified based on inventory requirements and (iii) strategies to address the hotspot can be directed towards reducing the consumption of inventory items contributing the most to the CO₂ footprint of CCU concrete. 19 out the 23 studies (A[1] + C[10] + D[6] + E[2]) report primary experimental data. However, among the 19 studies, there is no study that accounts for data uncertainty, only 1 study accounts for change in compressive strength (A[1]) and 3 studies (A[1]+E[2]) account for the CO₂ impact of capture, transport and utilization of CO₂.*
- *CO₂ sequestered through concrete curing or mixing: If a study quantifies the climate benefit based on only the CO₂ sequestered, it is not accounting for the CO₂ impact from the change in compressive strength and the upstream processes related to capture, transport and utilization of CO₂. The inclusion of the CO₂ impact of the upstream processes decreases the net CO₂ benefit of CCU concrete.*

As seen in Figure S2, 17 studies account for CO₂ sequestered (A[1] + B[4] + C[10] + E[2]). Only 2 (E[2]) of the 17 studies account for the CO₂ impact of capturing, transporting and utilizing CO₂. None of the 17 studies account for data uncertainty, change in compressive strength of CCU concrete, compensation of the energy penalty of CO₂ capture or allocation of CO₂ emissions to the SCMs.

Due to the above knowledge gaps and methodological shortcomings, the findings from one study cannot be generalized to other CCU studies and there is no comprehensive assessment to generalize and support claims on the net CO₂ impact of CCU concrete. We further illustrate the shortcomings through three examples from the reviewed literature

- 1. Example 1 - Study number 14 in region A of the Venn diagram (Monkman, S., & MacDonald, M., 2017)³: The study uses point values for material and energy inventory items and does not account for the variability in inventory data and uncertainty in life cycle data when quantifying the net CO₂ benefit of CCU concrete. Moreover, this study investigates CO₂ mixing in concrete and, therefore, the findings are not applicable to CO₂ curing of concrete. In addition, the study does not account for the (i) allocation of the CO₂ emissions between the supplementary cementitious materials (SCMs) which are by-products of coal electricity and steel production and the (ii) compensation of the energy penalty associated with carbon capture.*
- 2. Examples 2 and 3 - Study number 21 (Sanna et al)⁴³ and study number 20 (Hepburn et al)⁴⁴ in region B of the Venn diagram: The 2 studies quantify the CO₂ benefit based on the CO₂ sequestered in concrete and do not account for the other 4 aspects in Figure S2. As a result, the 2 studies do not quantify the net CO₂ benefit of CCU concrete production on a life cycle basis.*

The authors use previous studies as the basis of their analysis but do not compare the insights from those papers with their own to demonstrate that the current analysis is more robust and/or novel. Three examples of work that should be considered and discussed in the paper are:

1. Monkman and MacDonald., 2017, Cleaner Production, Abstract/Table 6
2. Sanna et al., 2014, Chemical Society Review, Table 7
3. Hepburn et al., 2019, Nature, Figure 3

While two of these papers are included in the reference list, they are not compared to explain what insights are being contributed from this paper that are more robust and/or novel.

Response: As explained in the response to the previous comment, we have specifically included and discussed the shortcomings of the three studies in the literature review as illustrative examples (SI Section 13). References 14, 20, 21 in the excel contained in SI Section 13 (<https://rb.gy/afq1te>) refer to the above 3 studies.

- I applaud the authors for attempting to treat uncertainty using a more sophisticated technique beyond the standard Monte Carlo commonly used in the LCA literature. However, aggregating the range of inputs to generate generic distributions seems like a big limitation in this study. Common inputs that lack the context of the studies themselves seem problematic. The context of each study should be considered and unique factors considered in the determination of distributions. If not, some support for this simplification should be provided.

Response: We have accounted for the range of the material and energy data required to produce CCU and conventional concrete based on the values reported in experiments in each dataset.

The range is individually determined for the following CCU and conventional concrete parameters from each dataset– compressive strength, OPC used, fine aggregate used, coarse aggregate used, water used, SCM used, steam curing hours. In addition, for CCU concrete, we obtain the CO₂ curing hours and CO₂ utilized for each dataset individually.

The SI provides a detailed report of the literature review and the material and energy inventory data used in CCU and conventional concrete production for each of the 99 datasets (SI Section 2) and CO₂ capture, transportation and utilization (SI Section 3). For example, the inventory items from each of the 99 datasets is listed in the excel sheet available here: <https://www.dropbox.com/s/ueex0uk3m95q3eo/Literature.xlsx?dl=0>

As a result, we retain the context and unique factors of CCU and conventional concrete production reported in each of the 99 datasets when determining the distribution of the CO₂ impact of CCU and conventional concrete and avoid combining inventory data across different datasets.

The only common factors of analysis across the datasets are the life cycle CO₂ characterization factors (e.g. the CO₂ intensity of the cement production (948 g CO₂/kg OPC), which are obtained from the Ecoinvent database (e.g. SI Table 1).

Moreover, the sensitivity analysis quantifies the sensitivity of the net CO₂ benefit of CCU concrete to the uncertainty in the underlying data (Figure 5 in main paper).

Also, are 2000 runs in the Monte Carlo Simulation sufficient to generate stable results? This should be discussed and justified as highly uncertain systems typically require 10,000 or more runs to do this.

Response: We have increased the number of runs in the Monte Carlo simulations to 10,000

- The consideration of strength is important. This goes beyond previous LCAs but still does not go far enough to provide insights about the potential use of these materials. There are issues around practicality of application, lifetime that will also impact the viability and potential of these technologies.

Lastly, there are issues related not just to risks but also to perception and risk tolerance and differences in different geographies, contexts that are important to note if not deal with in a quantitative way.

Response: We have text in the revised manuscript to discuss issues beyond compressive strength. We discuss the lifetime and geography specific factors which impact the practicality of the CCU concrete application.

Lines 356-366

“The impact of CO₂ curing or CO₂ mixing on the lifetime behaviors and geography specific factors are important practical considerations when using CCU concrete in commercial applications. The lifetime behaviors are impacted by the variations in the geographical sources of raw materials, mix type, product type and the service environment in which the concrete is deployed. For example, the impact of CO₂ curing on concrete lifespan would differ substantially

between concrete with and without steel reinforcement, due to the heightened steel corrosion⁴⁸ caused by the CO₂-induced pH reduction.⁴⁹ Additionally, geography specific factors such as sulfate-rich soils,⁵⁰ cold regions,³⁹ or acidic environments²² can impact the durability of CCU concrete. The findings of this study can be further complemented by future research which quantifies the impact of variations in the lifetime and the geography specific factors on the net CO₂ benefit of CCU concrete.”

In addition, the manuscript contains text about the importance of considering the durability of CCU concrete.

Lines 306-325

“Improve understanding on the impact of CO₂ curing on durability: The findings of this research are based on the compressive strength property of CCU concrete, which is limited from a lifecycle perspective. Prior studies show that construction and repair frequencies are key drivers in determining concrete life cycle CO₂ impacts.^{36,37} Therefore, the effect of CCU on concrete durability must be considered when analyzing life cycle CO₂ emissions. Preliminary lab scale studies demonstrate that CO₂ curing improves durability related parameters such as permeability, sorptivity and sulfate and acid resistance.^{22,38-40} However, the variability in the curing conditions and the design mixes used in the studies should be accounted for to ensure that durability gains can be consistently realized when CO₂ curing of concrete is adopted at a commercial scale. Future work can prioritize standardizing the CO₂ curing protocol (e.g. the steam curing time, pre-hydration time, post-hydration time) and study the resulting durability impact on different design mixes (e.g. use of different SCMs), with the overall goal of identifying optimal curing conditions and design mixes to maximize durability. This applies to ready-mix concrete and general precast applications with end-products such as masonry units,^{32,35} pipes²² and pavers.⁴¹ In addition, CO₂ curing for reinforced concrete needs further investigation due to

the possibility of increased risk of steel reinforcement corrosion led by concrete carbonation.

Moreover, CCU can be potentially synergized with established strategies for concrete crack width control, e.g., engineered cementitious composites with microfiber reinforcement, to further promoting concrete durability.^{42,43}”

Minor Issues

- There are issues with referencing that concerns me in terms of attention to detail throughout the paper. For example, reference 38 is incomplete.

Response: We have taken care to address the errors in the references.

- The title is not clear nor descriptive of the research presented. Is CCU for concrete not worth doing? It seems like the answer is more nuanced than this and should be part of the paper.

Response: There is a 15-word limit on the title of the paper. Given this limit, we wanted to communicate the core aspects of the research and findings contained in this manuscript

- CO₂ is utilized through two key processes - curing and mixing - of CCU concrete
- Based on the current state of the art, CCU concrete is more likely not to produce a climate benefit (i.e. net CO₂ benefit is negative or increase in net CO₂ emissions in 56-68 of 99 studies). The words ‘might not’ account for the probabilistic nature of the analysis and the findings

As a result, we have modified the title to be *“Carbon dioxide utilization in concrete curing or mixing might not produce a net climate benefit”*

To improve nuance and discuss how CCU concrete is worth doing from a climate perspective, we include a section on strategies to increase the net CO₂ benefit of CCU concrete based on the findings of the research. Please refer the section titled *“Strategies to Improve the Net CO₂ Benefit of CCU Concrete”* on Line 286

- Abstract: “existing estimates do not account for the life cycle CO₂ impact from the capture, transport and utilization of CO₂”. I don’t agree with this statement. There are many LCA studies on this topic that have been published that take these stages into account.

Response: In the modified abstract, we have qualified this statement with additional knowledge gaps in existing literature, which studies do not account for when determining the climate benefit of CCU concrete. This includes

- CO₂ impact from the capture, transport and utilization of CO₂,
- change in compressive strength in CCU concrete
- uncertainty and variability in CCU production conditions

As discussed in the response to the previous review comment on the novelty contained in this study, we review 23 studies (SI Section 13 and demonstrate that no previous study has accounted for the knowledge gaps and this prevents a robust and comprehensive assessment of the net CO₂ impact of CCU concrete.

- No mention of methods in the abstract related to how CO₂ is being incorporated.

Response: We have included text in the abstract to address this comment. CO₂ is used in the curing and mixing of concrete

Line 17-18

“By accounting for these factors, we determine the net CO₂ benefit when CCU concrete produced from CO₂ curing and mixing substitutes for conventional concrete.”

- Line 33: What unit basis is this comparison occurring and what are the other two products?

Response: We have modified the text in the revised manuscript to address this review comment

Lines 32-34

“Concrete along with aggregates and chemicals/fuels are end-products with the potential to sequester the maximum quantity of CO₂ (in gigatons).^{1,3}”

What type of mineralization process is discussed here?

Response: We have included text in the Introduction section to address this review comment

Lines 38-45

“However, this study focuses on the two approaches of CO₂ mixing and CO₂ curing as they are more extensively analyzed and applied for CO₂ utilization in concrete (supplementary information (SI) Section 2). In CO₂ mixing, high-purity CO₂ is injected into fresh concrete during batching and mixing. The CO₂ binds to the calcium silicate clinker in OPC to form nano-scale CaCO₃ particles.^{15,16} In CO₂ curing, CO₂ is utilized as a curing agent⁵ to accelerate precast

concrete fabrication. A review of CO₂ curing and mixing studies reveals that the CO₂ uptake potential in CO₂ curing of precast concrete applications is significantly higher than in CO₂ mixing (SI Figure S6)."

• Line 38: Did your findings support this? As written, it sounds like you are endorsing this estimate while your analysis seems to conclude something different.

Response: This observation is based on the literature review we conducted. Please refer Figure S6 in the SI where we plot the CO₂ utilized in curing and mixing datasets.

We have modified the text in revised manuscript to make this claim clear

Lines 43-45

"A review of CO₂ curing and mixing studies reveals that the CO₂ uptake potential in CO₂ curing of precast concrete applications is significantly higher than in CO₂ mixing (SI Figure S6)."

• Line 55: There is only benefit to representing the uncertainty and variability accurately. It is possible to do more harm than good if the input values used to inform the distributions are based on guesses or a combination of datasets without appropriately combining these datasets.

Response: We account for the variability and uncertainty in the material and energy data required to produce CCU and conventional concrete based on the values reported in experiments in each dataset. The inventory items which are individually determined for CCU and conventional concrete in each dataset include – OPC used, fine aggregate used, coarse aggregate used, water used, SCM used, steam curing hours. In addition, for CCU concrete, we determine the CO₂ curing hours and CO₂ utilized for each dataset individually. The values are included in the excel linked in section 2 of the SI

<https://www.dropbox.com/s/ueex0uk3m95q3eo/Literature.xlsx?dl=0>

The SI provides a detailed report of the literature review and the material and energy inventory data used in CCU and conventional concrete (SI Section 2) and CO₂ capture, transportation and utilization (SI Section 3). The life cycle factors used characterize the CO₂ impact of the material and energy inventory data are obtained from Ecoinvent and reported in Table 1 of SI.

As a result, we have taken care to avoid guesses.

Moreover, we quantify the CO₂ impact of CCU and conventional concrete production individually for each of the 99 datasets based on the material and energy inventory data reported in each dataset. As a result, we avoid combining inventory data across different datasets and retain the production conditions of production of CCU and conventional concrete in each of the 99 datasets.

The only common factors of analysis across the datasets are the life cycle CO₂ characterization factors (e.g. the CO₂ intensity of the cement (948 g CO₂/kg OPC), which are obtained from the Ecoinvent database (e.g. SI Table 1).

Moreover, the sensitivity analysis quantifies the sensitivity of the net CO₂ benefit of CCU concrete to the uncertainty in data (Figure 5 in main paper).

• Line 59: how the accounting for uncertainty helps identify hotspots and increase the CO₂ benefits should be made more explicit.

Response: We have addressed this comment by including this text in the revised manuscript

Lines 74-77

“An uncertainty assessment in the early stages of technology development can determine process parameters and inventory items that are the most significant

contributors to the CO₂ burden of CCU concrete and, thereby, help identify research strategies that are most effective in addressing the hotspots.”

• Line 64: What are the 13 upstream processes?

Response: The 13 upstream processes include the production of materials required to produce CCU and conventional concrete and capture, transport and utilize CO₂

We have communicated this in Lines 81-83

“The net CO₂ benefit accounts for the life cycle CO₂ impact of the 13 upstream processes to capture, transport and utilize CO₂ and produce and transport the materials used in concrete.”

Furthermore, we have depicted and numbered the processes in Figure 1 in the main paper.

A detailed explanation of the 13 processes and how the CO₂ footprint of each is determined is provided in the methods section (Lines 421-548)

• Line 65: How did you ensure that the stages were consistent in the boundary? Please describe separately in methods section.

Response: We have explained how we maintain consistency in the section on Functional Unit in the Methods Section

Lines 401-417

“The use of CO₂ during mixing or curing changes the compressive strength of CCU concrete when compared to concrete produced through conventional mixing or curing. In addition, an energy penalty (E_p kWh) is incurred for CCU concrete in power plants due to the energy associated with capturing the CO₂, which is used in the curing or mixing of CCU concrete (φ_{CCU} , kg CO₂). E_p is not incurred when conventional concrete is produced since there is no CO₂ capture. Therefore, the net CO₂ benefit of substituting CCU concrete for conventional concrete should account for the CO₂ impact from the change in the compressive strength and E_p , which is incurred in power plants only when CO₂ is captured.

As a result, we use a functional unit of concrete with 1 MPa compressive strength and 1 m³ of volume and E_p kWh of electricity.

The functional unit accounts for the change in compressive strength and ensures consistency by normalizing the materials and energy consumed for producing 1 m³ of CCU and conventional concrete to 1 MPa of compressive strength. The inclusion of E_p kWh of electricity in the functional unit accounts for the difference in CO₂ emissions from electricity generation without CO₂ capture in the conventional concrete pathway and with CO₂ capture in the CCU concrete pathway. E_p is determined based on the mass of CO₂ captured from the power plant (SI Table SI Process 8).”

- Line 74: Why are you describing results in the introduction section?

Response: We have removed the results from the introduction section

- Line 100: Sentence structure, “...which signifies that the in the 2000....”.

Response: We modified the text to address the review comment

Lines 111-112

“For dataset 1 the likelihood is 0%, which signifies that TOT_{CCU} is greater than TOT_{Conv} in all of the 10000 Monte Carlo runs.”

- Line 144: How much has the strength decreased by? A percentage would be very helpful in interpreting the impacts.

Response: We have included text to address this review comment

Lines 155-161

“The finding from the sensitivity analysis for dataset 1 can be attributed to the compressive strength of CCU concrete (16-17.4 MPa) being lower than conventional concrete (18 – 18.6 MPa). The mean compressive strength of CCU concrete (16.7 MPa) is 9% lower than that of conventional concrete (18.3 MPa). This implies that in dataset 1 a greater mass of OPC is

produced for CCU concrete to achieve the same compressive strength as conventional concrete. In dataset 1, the OPC produced per MPa for CCU concrete is 24.8 kg/MPa and for conventional concrete is 22.6 kg/MPa.”

- Line 146: How much OPC is added/substituted to compensate?

Response: We modified the text to address the review comment

Lines 158-161

“This implies that in dataset 1 a greater mass of OPC is produced for CCU concrete to achieve the same compressive strength as conventional concrete. In dataset 1, the OPC produced per MPa for CCU concrete is 24.8 kg/MPa and for conventional concrete is 22.6 kg/MPa.”

- Line 147: What is the final strength of the CCU and conventional concretes?

Response: We have included text to address this review comment

Lines 155-158

“The finding from the sensitivity analysis for dataset 1 can be attributed to the compressive strength of CCU concrete (16-17.4 MPa) being lower than conventional concrete (18 – 18.6 MPa). The mean compressive strength of CCU concrete (16.7 MPa) is 9% lower than that of conventional concrete (18.3 MPa).”

- Line 149: How much CO₂ is captured and used in curing vs. what is increased from OPC addition?

Response: We have included text to address this review comment

Lines 162-168

“Using a mean value of 0.948 kg CO₂/kg OPC for the life cycle CO₂ footprint of OPC (SI Table S2), the CO₂ emissions from OPC production for CCU concrete is 23.5 kg CO₂/MPa and for conventional concrete is 21.4 kg CO₂/MPa. Therefore, the difference between the CO₂ emissions from OPC production for CCU and conventional concrete is 2.1 kg CO₂/MPa, which is greater

than the CO₂ utilized in CCU concrete (1.1 kg CO₂/MPa, SI Section 1). As a result, the life cycle CO₂ emissions from the increased OPC production in CCU concrete is greater than the CO₂ utilized for curing the CCU concrete.”

- Line 171: What is the source/type of energy?

Response: We modified the text to address the review comment and refer to the electricity used in CO₂ curing.

Lines 189-191

“Additionally, in category 1 datasets, the electricity use in the CO₂ curing process is the second key contributor to the increase in the total life cycle CO₂ emissions from CCU concrete (e.g. datasets 46, 47 and 48 in Figure 5).”

- Line 223: Why is allocation used? Discuss the rationale.

Response: We have modified text in the revised manuscript to provide a rationale for allocation

Lines 236-242

“Two types of SCMs are used in concrete production - ground granulated blast furnace slag and fly ash. Slag is a by-product of pig-iron production and fly ash is a by-product of electricity generation in coal power plants. As a result, we need a method to allocate total CO₂ emissions between slag and pig-iron and between fly-ash and coal electricity. We use three methods - system expansion (SE), mass-based (MA) and economic value-based allocation (EA) - to allocate and account for the life cycle CO₂ emissions from producing the SCMs (process P5, SI Sections 4, 5 and 6).”

- Line 289: What is the application and end-product of your study? What does your study about what applications should be recommended moving forward?

Response: Our study presents a general approach which quantifies the net CO₂ benefit based on CCU concrete offsetting conventional concrete and is not restricted to a specific application of CCU concrete.

However, we have modified the text in this paragraph to identify potential applications for which the findings are applicable.

Lines 316-321

“Future work can prioritize standardizing the CO₂ curing protocol (e.g. the steam curing time, pre-hydration time, post-hydration time) and study the resulting durability impact on different design mixes (e.g. use of different SCMs), with the overall goal of identifying optimal curing conditions and design mixes to maximize durability. This applies to ready-mix concrete and general precast applications with end-products such as masonry units,^{32,35} pipes²² and pavers.⁴¹”

- Line 296: Keep tense consistent throughout the paper.

Response: We have taken care to use the past tense consistently throughout the revised manuscript.

Lines 369-371

“We conducted a literature review to obtain the 99 datasets from 19 studies presenting life cycle material and energy inventory data and process parameters for the production of CCU and conventional concrete.”

- Line 298: How were these studies located and selected? Is this comprehensive?

Response: We identified the studies based on three criteria, which is explained in the manuscript.

Lines 371-379

“The literature review identified the 19 studies^{16,19,22,23,31-33,35,38,40,51-59} as they were the only ones to report the following three items (i) the design mix consisting of the energy and material inventory required for the production of conventional and CCU concrete (SI Section 2). The energy and material inventory are required to determine the life cycle CO₂ impact of producing conventional and CCU concrete; (ii) the quantity of CO₂ used in mixing or curing of concrete.

This is required to determine the life cycle CO₂ impact of capturing, transporting and utilizing the CO₂ used in producing CCU concrete; and (iii) the compressive strength of CCU and conventional concrete at the end of 28 days, which helps account for the change in the material property between conventional and CCU concrete.”

• Line 307: What is the compressive strength you selected? How? And does it differ for each of the 99 datasets?

Response: The 28-day compressive strength of CCU and conventional concrete are obtained from each of the 99 datasets, which were identified through the literature review.

The values of the compressive strength of the 99 datasets are reported in the excel sheet linked in SI Section 2.

(<https://www.dropbox.com/s/ueex0uk3m95q3eo/Literature.xlsx?dl=0>)

The compressive strength differs for each of the 99 datasets.

We have plotted the compressive strength of the CCU and conventional concrete and the corresponding differences in compressive strength from the 99 datasets in Figure S2 and Figure S3 in the SI.

• Line 338-340: Sentence not complete.

Response: We modified the text to address the review comment

Lines 413-416

“The inclusion of E_p kWh of electricity in the functional unit accounts for the difference in CO₂ emissions from electricity generation without CO₂ capture in the conventional concrete pathway and with CO₂ capture in the CCU concrete pathway.”

• Line 352: Delete one “in”

Response: We modified the text to address the review comment

Lines 424-425

“The expression used to determine the total life cycle CO₂ emissions from producing CCU concrete based on the CO₂ emissions from the 13 processes is presented in Equation 1.”

• Line 437: Awkward sentence: What is the difference between tanker and heavy-duty truck?

Response: We have revised the manuscript to consistently use the term “Semi-trailer truck” based on the specifications of commercial truck manufacturers.

Lines 516-519

“Process 9 - CO₂ Transportation: This analysis assumes that the captured CO₂ is transported in a semi-trailer truck (SI Section 3 “CO₂ Transportation”) as it is necessary to supply the CO₂ from the site of capture to geographically disperse concrete curing or mixing facilities, which are primarily accessible by road.²¹”

• Line 441: “and the CO₂” font is different.

Response: We have changed the font in the revised manuscript

Lines 519-522

“The CO₂ emissions from transporting CO₂ is the product of the total weight (ϕ_{CCU} plus the tare weight), the distance over which the transport occurs (D_{CO_2} in km) and the CO₂ intensity of transportation emissions of a semi-trailer truck ($\phi_T = 112$ g CO₂ per ton km, SI Table S11).”

• Line 441/442: change to “...the CO₂ intensity of transportation emissions of a heavy-duty truck.”

We modified the text to address the review comment

Lines 521

“...and the CO₂ intensity of transportation emissions of a semi-trailer truck.”

Reviewer 3

This paper presents an analysis for identifying the cost-benefit analysis of CO₂ sequestration in concrete production. Considerable research and development are currently ongoing to investigate the possible use of CO₂ products in concrete production. The work presented is, therefore, relevant and of interest to the readers. The article is also well-written. ***The reviewer suggests that the paper might be accepted for publication pending addressing the reviewer's comments provided below.***

Response: We thank the reviewer for the constructive feedback as it helps to better contextualize our findings and claims based on the two approaches of CO₂ utilization (curing and mixing) analyzed in this study and improve the articulation of the limitations of the study. A detailed response to each of the review comments is provided below.

1. The authors need to provide a range of concrete mix variables and concrete characteristics used in the study. For instance, it is crucial to provide a table explaining the range of cement content kg/m³, SCM/cement ratio and SCM kg/m³, water/cement ratio, and concrete compressive strength range and concrete densities considered in the study.

Response: We have modified the SI in the revised manuscript to include figures presenting the above ranges. Please refer Figures S1 to Figure S5 in Section 2 in the SI.

To inform the reader, we have referred to this in the main manuscript

“The findings from this analysis are based on the design mixes, material usage, compressive strength and parameters such as the CO₂ curing duration, water to cement and SCM to cement ratios obtained from the 99 datasets (SI Section 2 Figures S1 to S5).”

The authors might then make their discussion while pointing out that the analysis they conducted is limited to this range of variables and characteristics. It is essential to indicate that this conclusion is not general and might be affected by changes in these variables. This addition will also provide useful information to other researchers to examine which of those variables (in concrete mix) or characteristics (e.g., strength as indicated by the authors) might dictate the net benefit of using CO₂ in concrete.

Response: We have included a new paragraph in the revised manuscript to address this review comment (Lines 280-285).

“The findings from this analysis are based on the design mixes, material usage, compressive strength and parameters such as the CO₂ curing duration, water to cement and SCM to cement

ratios obtained from the 99 datasets (SI Section 2 Figures S1 to S5). It is important to note that the findings do not preclude future research (as discussed below) from optimizing the design mixes, curing processes and material properties to increase the net CO₂ benefit from CCU concrete.”

Furthermore, we include a section on potential research strategies to improve the net CO₂ benefit from CCU concrete (Lines 286).

2. It is also critical to point out the limited technologies considered to include CO₂ in concrete. The data presented by the authors are limited to two techniques where CO₂ was used in fresh concrete or concrete was cured using CO₂. Alternative methods for incorporating CO₂ in concrete have been reported in the literature and used in the industry (e.g., dispersing CO₂ particles in fresh concrete). While it is not expected that the authors will cover all those methods, the authors need to acknowledge this limitation in their introduction and when discussing their results. The conclusion made is strongly related to the limited techniques used for incorporating CO₂ in concrete.

Response: We have included text in the Introduction section (Lines 35-49) and the section following the “Results” (Lines 335-346) to address this review comment. We provide an overview of alternate approaches to utilize CO₂ in concrete production and mention how we assess two widely used approaches – CO₂ curing and CO₂ mixing – in this paper.

Lines 35-40

“Multiple emerging approaches such as carbonation of recycled concrete aggregates,⁹ CO₂ sequestration in alternative MgO based binders,¹⁰ CO₂ mineralization in industrial waste-derived aggregates and fillers^{11,12} and CO₂ dissolution in mixing water^{13,14} have been investigated for CO₂ utilization in concrete. However, this study focuses on the two approaches of CO₂ mixing and CO₂ curing as they are more extensively analyzed and applied for CO₂ utilization in concrete (supplementary information (SI) Section 2).”

Lines 344-355

“This analysis focusses on the use of pure CO₂ and two approaches of CO₂ utilization - curing and mixing - as they are more extensively investigated (e.g. 99 datasets used in this study) than alternate approaches such as concrete curing with flue gas,^{21,45-47} carbonation of recycled concrete aggregates,⁹ CO₂ sequestration in alternative MgO based binders,¹⁰ and CO₂ dissolution in mixing water.^{13,14} The increased availability of experimental data is necessary to robustly quantify the net CO₂ benefit of CCU concrete and account for the impact of data uncertainty and process variability on the results. For example, further experimental research can generate data on the variability in the compressive strength of flue-gas cured concrete properties when the 2-week curing time is reduced⁴⁷ and the CO₂ concentration in the flue gas is varied.⁴⁵ With increased availability of inventory and process data from future experimental research, the life cycle approach presented in this study can be extended to quantify the net CO₂ benefit of CCU concrete produced from flue gas and other alternate approaches of CO₂ utilization.”

3. There is a definite climate benefit on sequestering CO₂ from the climate into products; the question the authors are answering is whether the net benefit is positive or negative. The discussion and conclusion sections, as well as the title, shall point out to the fact the paper is discussing “net benefit” not “benefit.” For instance, in line 20 in the abstract, “benefit” shall be changed to “net benefit.” This issue is overlooked in multiple locations throughout the paper and shall be fixed.

Response: We have taken care to use the term “net lifecycle CO₂ emissions” in the abstract.

In the main paper, we define (Lines 78-87) and, thereon, consistently use the term “net CO₂ benefit”.

4. Given the above comments, it is the reviewer’s opinion that the title is overstated and might be a bit misleading. The title indicates that the work covers all types of concretes and all methods for incorporating CO₂ in concrete, and this is not true. The reviewer suggests that the authors shall change the title to indicate this is a preliminary or limited set analysis and that the benefit considered herein is the net benefit something like “Preliminary analysis indicates carbon utilization in concrete production may or may not produce a net climate benefit”.

Response: We have revised the title and the manuscript at various sections and addressed this review comment as follows

- We have modified the title to “*Carbon dioxide utilization in concrete curing and mixing might not produce a net climate benefit*”. This helps communicate that the findings from this study is limited to two approaches of producing CCU concrete - CO₂ curing and CO₂ mixing. As recommended by the reviewer, we include the word “net” in the title to communicate that we are quantifying the CO₂ benefit on a net basis.
- As explained in the response to review comment 2, we have included text in the “Introduction” section (Lines 35-40) to explain that our analysis focuses on - CO₂ curing and CO₂ mixing.
- As explained in the response to review comment 2, we have included text in the section following the results (Lines 344-355) to explain that findings are based on the CO₂ curing and CO₂ mixing approaches of CCU concrete production.

REVIEWERS' COMMENTS

Reviewer #3 (Remarks to the Author):

The authors addressed all the comments by the reviewers satisfactorily.

REVIEWER COMMENTS

Reviewer #3 (Remarks to the Author):

The authors addressed all the comments by the reviewers satisfactorily.

Response: We thank the reviewer for the positive feedback.